

# Development and validation of a new MODIS snow-cover-extent product over China

Xiaohua Hao[1,2], Guanghui Huang[3,1,2], Zhaojun Zheng[4,5], Xingliang Sun[1,6], Wenzheng Ji[1], Hongyu Zhao[1], Jian Wang[1,2], Hongyi Li[1,2], Xiaoyan Wang[3]

[1]Heihe Remote Sensing Experimental Research Station, Northwest Institute of Eco-Environment and Resources, Chinese Academy of Sciences, Lanzhou 730000, China
[2]Key Laboratory of Remote Sensing of Gansu Province, Northwest Institute of Eco-Environment and Resources, Chinese Academy of Sciences, Lanzhou 730000, China
[3]College of Earth and Environmental Sciences, Lanzhou University, Lanzhou 730000, China
[4]National Satellite Meteorological Center, China Meteorological Administration, Beijing 100081, China
[5]Key Laboratory of Radiometric Calibration and Validation for Environmental satellites, China Meteorological Administration, Beijing 100081, China
[6]Engineering Laboratory for National Geographic State Monitoring, Lanzhou Jiaotong University, Lanzhou 730070, China

*Correspondence to*: Guanghui Huang (huanggh@lzu.edu.cn)

**Abstract.** Based on the MOD09GA/MYD09GA 500-m surface reflectance, a new MODIS snow-cover-extent (SCE) product over China has been produced by the Northwest Institute of Eco-Environment and Resources (NIEER), Chinese Academy of Sciences. The NIEER MODIS SCE product contains two preliminary clear-sky SCE datasets — Terra-MODIS and Aqua-MODIS SCE datasets, and a final daily cloud-gap-filled (CGF) SCE dataset. The formers are generated mainly through optimizing snow-cover discriminating rules over different land-cover types, and the latter is produced after a series of gap-filling processes such as aggregating the two preliminary datasets, reducing cloud gaps with adjacent information in space and time, and eliminating all gaps with auxiliary data. Validation against 362 China Meteorological Administration (CMA) stations shows during snow seasons the overall accuracies (OA) of the three datasets are all larger than 93%, the omission errors (OE) are all constrained within 9%, and the commission errors (CE) are all constrained within 10%. Biases ranging from the lowest 0.98 to the medium 1.02, to the largest 1.03 demonstrate on a whole the SCEs given by the new product are neither overestimated nor underestimated significantly. Based on the same ground reference data, we found the new product's accuracies are clearly higher than those of standard MODIS snow products, especially for Aqua-MODIS and CGF SCE. For examples, compared with the CE of 23.78% that the standard MYD10A1 product shows, the CE of the new Aqua-MODIS SCE dataset is 6.78%; the OA of the new CGF SCE dataset is up to 93.15%, versus 89.54% of the standard MOD10A1F product and 84.36% of the standard MYD10A1F product. Besides, as expected snow discrimination in forest areas is also improved significantly. An isolated validation at four forest CMA stations demonstrates the OA has increased by 3 – 10 percentage points, the OE has dropped by 1 – 8 percentage points, and the CE has dropped by 4 – 21 percentage points. Therefore, our product has virtually provided more reliable snow knowledge over China, and thereby can better serve for hydrological, climatic, environmental, and other related studies there.



# 1 Introduction

Snow is one of the most active elements on the land surface. Because of its unique properties of high shortwave reflectivity, low longwave emissivity, and high phase-change latent heat, its covering may significantly alter the surface radiation budget, exchanges of energy and moisture between the atmosphere and the surface, and thereby our climate and weather systems (Henderson et al., 2018; Huang et al., 2019; Warren, 1982). On the other hand, seasonal snow is an important supply source, providing precious freshwater for many arid and semi-arid regions (Li et al., 2019). Snow-cover-extent (SCE), therefore, is

an indispensable parameter in climatic, weather, hydrological, environmental, and other related studies.

With the development and progress of space technology, satellite remote sensing has become a primary option to monitor snow-cover conditions on the Earth (Frei et al., 2012). In particular, as one of the most successful satellite missions, since launched in 1999 the Moderate Resolution Imaging Spectroradiometer (MODIS) has been widely used to acquire SCE information from regional to global scales (Gafurov, et al., 2016; Hall and Riggs, 2007; Hall et al., 2002; Muhammad and

Thapa, 2020). The reasons for this arise from: 1) MODIS is a multi-spectral sensor with a specific snow detecting band around 1.64 μm; 2) features of nearly global coverage, double-satellite observations (Terra and Aqua), and suitable spatio-temporal resolutions, also make it reasonable to acquire regional and global SCE.

The National Snow and Ice Data Center (NSIDC) routinely produces and continually updates the standard MODIS snow products, in which the MOD10A1 and MYD10A1 provide conventional SCE information under clear skies, while the

MOD10A1F and MYD10A1F are the recently released cloud-gap-filled (CGF) products (Hall et al., 2019). In standard MOD10A1 and MYD10A1 products, the core is datasets of the Normalized Difference Snow Index (NDSI) that is defined as,

$$NDSI = (b4 − b6)/(b4 + b6), \tag{1}$$

where b4 and b6 represent the MODIS band 4 (0.55 μm) and band 6 (1.6 μm) reflectance, respectively. For clear skies, NDSI is generally effective to distinguish snow cover or not because the snow has a distinct NDSI characteristic relative to other

common land covers. The previous products adopted NDSI = 0.4 as the threshold to discriminate snow-cover pixels and produced binary snow-cover datasets (Riggs et al., 2006), whereas the newest C6 products only provide NDSI datasets, but advise NDSI = 0.1 as the new threshold (Riggs et al., 2016).

While standard MODIS products have already been applied widely in various research (Li et al., 2019; Mcguire et al., 2006; Rodell and Houser, 2004), there are several appreciable shortcomings. First, several studies (Hall and Riggs, 2007; Hao et al.,

2019; Zhang et al., 2019) have shown that the optimal NDSI threshold may vary with land-cover types, and using a fixed value for global products, regardless of 0.4 or 0.1, may lead to considerable uncertainty. Second, standard MOD10A1 and MYD10A1 products often exhibit large errors in forest areas (Maurer et al., 2003; Parajka et al., 2012; Poon and Valeo, 2006). For example, an evaluation conducted by Chen et al. (2014) indicates in many forest regions of Northeast China the omission error or commission error may be up to 40%. Last but not the least, for snow discrimination theoretically using

surface reflectance, is also more reasonable than using top-of-atmosphere (TOA) reflectance (what the standard products do), as the atmospheric contribution may virtually reduce the NDSI contrasts between snows and other land covers.

Despite this, what the previous standard products (before C6) were criticized most is they cannot provide complete SCE knowledge due to abundant cloud-induced gaps (Liang et al., 2008). However, as mentioned before during the preparation of our paper, two new cloud-free products, MOD10A1F and MYD10A1F, were released in the newest C6.1 products (but as of

now, only parts of years are available). Although this will promote MODIS standard products' applications dramatically, there are still several deficiencies, as far as our experience. First, because of MODIS tradition, they are produced for Terra-MODIS and Aqua-MODIS separately, but many studies (Huang et al., 2016; Gao et al., 2010; Gafurov and Bardossy, 2009) have revealed that their synergy may be a better way to remove cloud influence. Second, their CGF mapping algorithm virtually only replaces cloud gaps in the current day with the previous most-recent clear-sky observation (Hall et al., 2019).

Implications indicated by the subsequent clear-sky observation or spatially adjacent observations are all neglected. Under many scenarios, this seems unreasonable. For example, the snowfalls often happen on cloudy days, and if only the previous clear-sky observation is used, these days would be very likely mistaken for snow-free days.

Therefore, we decided to produce a new MODIS SCE product over China based on the Google Earth Engine (GEE) platform (Gorelick et al., 2017). Compared with the standard snow products, the following improvements are reached: 1) varying

NDSI thresholds with surface cover types are obtained by a volume of training data; 2) the approach that combines the Normalized Difference Vegetation Index (NDVI) and the Normalized Difference Forest Snow Index (NDFSI, Wang et al., 2015; Wang et al., 2020), is adopted to improve snow discrimination in forest areas; 3) NDSI is computed using surface reflectance instead of TOA reflectance; 4) a Hidden Markov Random Field (HMRF) based gap-filling technique (Huang et al., 2018) is used to reduce cloud-induced gaps, which can assimilate temporally and spatially adjacent information

simultaneously.

## 2 Data

### 2.1 MODIS products

Two kinds of MODIS products are used in our study chiefly. One is the standard surface reflectance products, including the MOD09GA and MYD09GA; the other is the composite land-cover-type product, namely the MCD12Q1. The MOD09GA

and MYD09GA provide us the main input, that is 500-m surface reflectance from MODIS band 1 to band 7, as well as mask information (e.g., cloud and water masks). The MCD12Q1 provides us the primary reference data — annual land-cover maps based International Geosphere Biosphere Program (IGBP) land cover classification system.

### 2.2 Landsat-8 OLI snow maps

The study involves two groups of Landsat-8 Operational Land Imager (OLI) snow maps across China during the 2013 –

2018 snow seasons (beginning on Nov. 1[st] through Mar. 31[st] of the next year), and each map contains three classes — snow, snow-free, and cloud. The first group is derived from 1509 scenes of OLI images and will be regarded as "true" values to





acquire the Terra-MODIS training data. The second group is derived from 1648 scenes of OLI images and will be used to acquire the Aqua-MODIS training data.

### 2.3 Auxiliary data

The auxiliary data we used include a snow-depth product generated from passive microwave satellite observations, a reanalysis land surface temperature (LST) product, and a Digital Elevation Model (DEM) product.

The snow-depth product is available at http://data.tpdc.ac.cn, providing daily and 0.25° snow-depth data over China from 1979 to 2020, which is generated by Che et al. (2008) and Dai et al. (2015) through a combination of multiple satellites' passive-microwave observations. The reanalysis LST product is available on the GEE, providing daily and 0.25° LST data

over China derived from ERA-5 global reanalysis (Munoz, 2019). The DEM product is one of the Shuttle Radar Topography Mission (SRTM) DEM products with a resolution of 90-m and directly accessible on the GEE, too. To match with MODIS data, all of the three products had been re-sampled or aggregated into 500-m.

### 2.4 Ground snow-depth measurements

Daily ground snow-depth observations up to 362 stations from the China Meteorological Administration (CMA) since 2000

will be used to validate or assess all associated MODIS SCE products in the study.

To ensure the qualities of the measurements, most CMA stations obey the following observing specifications: 1) snow-depth is measured manually in an open spot near the station using a professional ruler; 2) the measurements were conducted only when the fractional snow cover in the field of view is larger than 50%; 3) all observations were done at 8:00 Beijing time every morning. At each station, snow-cover condition is determined by the criterion proposed by Klein and Barnett (2003).

That is, if measured snow-depth is ≥ 1-cm, it is covered by snow; else it will be snow-free.

### 3 Method and product

### 3.1 Snow discrimination under clear-skies

To improve the snow discrimination under clear-skies, all decision rules used in the paper are re-adjusted according to a large number of training samples. As mentioned in Sect. 2.2, training samples of Terra-MODIS come from spatially and

temporally (in the same day) collocated MOD09GA surface reflectance with the first group of Landsat-8 OLI snow maps (within a MODIS pixel if 50% of OLI pixels are snow-covered, it will be deemed as a MODIS snow pixel, otherwise it will be a MODIS snow-free pixel); training samples of Aqua-MODIS come from MYD09GA surface reflectance collocated with the second group of Landsat-8 OLI snow maps. These training samples evenly distribute in major seasonal snow-cover regions across China. The former includes 21.20 million snow samples and 17.66 million snow-free samples totally, while

the latter includes 12.05 million snow samples and 12.65 million snow-free samples in total.



Note that it is necessary to train the decision rules for Terra-MODIS and Aqua-MODIS separately, as MYD09GA band 6 is not the directly observing data (many sensor's detectors of this band have broken since the Aqua launch), but the restored data using the algorithm of Wang et al. (2006).

### 3.1.1 Preliminarily screening

The purpose of the preliminarily screening is to preclude the pixels that are impossibly covered by snow completely. As done by the standard MODIS snow products (Riggs et al., 2006), the preliminarily screening involves the thresholds in MODIS band 2, band 4, and band 6, which will be re-adjusted in accordance with our training data.

Statistics on snow samples reveal for Terra-MODIS basically all of the snow samples (over 99%) are constrained in the condition of band 2 ≥ 0.15, band 4 ≥ 0.05, and band 6 ≤ 0.45, and for Aqua-MODIS basically all of the snow samples are
constrained in the condition of band 2 ≥ 0.12, band 4 ≥ 0.07, and band 6 ≤ 0.40.

Therefore, for preliminarily screening of snow, the Terra-MODIS decision rule is adjusted into that possible snow pixels should at least meet: band 2 ≥ 0.15, band 4 ≥ 0.05, and band 6 ≤ 0.45, and the Aqua-MODIS decision rule is adjusted into that possible snow pixels should at least meet: band 2 ≥ 0.12, band 4 ≥ 0.07, and band 6 ≤ 0.40.

### 3.1.2 Optimized NDSI thresholds

To obtain optimal NDSI thresholds for snow discriminations over different land cover types, all of the training samples are first grouped according to their land cover types that are indicated by the MCD12Q1 products.

With the type of "Barren or Sparsely Vegetated" as an example, the upper-left of Figure 1 presents the NDSI frequency distribution from Terra-MODIS training samples over this land cover, and the upper-right presents the fluctuation of overall accuracy (OA, see Sect. 4.1 for details) with different NDSI thresholds. From the figure, one can see around the cross-point
of snows and barren lands, the OA is highest (up to 98.34%) when the NDSI threshold = 0.08. Thus, 0.08 will be regarded as the optimal NDSI threshold for this land cover type. Similarly, the bottom-left and the bottom-right of Figure 1 present the NDSI frequency distribution and the fluctuation of OA derived from Aqua-MODIS training samples, respectively. It is apparent that for Aqua-MODIS the optimal NDSI threshold has changed to 0.06 and the highest OA is 95.20%.

Following the same philosophy, the optimal NDSI thresholds for the other seven land-cover types, such as "Grasslands",
"Croplands", "Urban and Built-up Lands", etc., are also obtained and listed in Table 1, together with their corresponding OAs. One can see that over these land-cover types the OAs are all larger than 95%, which demonstrates the effectiveness of the new thresholds.

Table 1 Optimal NDSI thresholds over eight non-forest land-cover types

| Land-cover types (IGBP) | NDSI thresholds (Terra) | Corresponding OA (%) | NDSI thresholds (Aqua) | Corresponding OA (%) |
|---|---|---|---|---|
| Barren or Sparsely Vegetated | 0.08 | 98.34 | 0.06 | 95.20 |





| Land-cover type | | | |
|---|---|---|---|
| Grasslands | 0.03 | 97.57 | -0.13 | 98.90 |
| Croplands | 0.17 | 98.89 | 0.26 | 99.02 |
| Urban and Built-up Lands | 0.17 | 98.44 | -0.12 | 99.02 |
| Cropland/Natural Vegetation | 0.21 | 99.06 | 0.00 | 98.00 |
| Closed Shrublands | 0.52 | 97.19 | 0.14 | 99.89 |
| Open Shrublands | 0.06 | 99.94 | 0.03 | 99.77 |
| Evergreen Broadleaf Forest | 0.41 | 99.95 | 0.40 | 99.30 |

However, as we expected, only using the NDIS criterion seems not accurate enough to discriminate snow cover over those

forest land-cover types, except the "Evergreen Broadleaf Forest" (due to its sparse distributions in China). For example, Figure 2 gives the Terra-MODIS NDSI frequency distribution and the fluctuation of OAs over the "Evergreen Needleleaf Forest". Obviously, here the commission error (CE, see Sect. 4.1 for details) is very large (> 35%), and the OAs drop severely (< 76%). Therefore, for the remaining seven forest land-cover types, we will adopt a new decision rule as elaborated next.

### 3.1.3 Optimized NDVI-NDFSI decision-rules for forests

The study of Wang et al. (2015) had showed compared with NDSI, their so-called Normalized Difference Forest Snow Index (NDFSI, namely using band 2 to substitute band 4 in Eq. (1)) may be better for snow discrimination in forest areas. Our preliminary investigation reinforces this, and the improvement is, in particular, evident when using NDFSI in conjunction with NDVI (Wang et al., 2020).

Following their studies, NDVI in our study is divided into seven segments, and the optimal NDFSI at each NDVI segment will be computed with the highest OA as the object (similar to the acquirements of optimal NDSI thresholds). For a given pixel, if computed NDFSI is ≥ the threshold in its corresponding NDVI segment, it will be regarded as a snow pixel; else it will be a snow-free pixel. With "Deciduous Broadleaf Forest" as an example, Figure 3 shows the NDVI-NDFSI number-density scatterplot that is derived from Terra-MODIS training samples over this land cover, as well as computed optimal

NDFSI thresholds at every NDVI segment. Here the OA of the new NDVI-NDFSI decision rule is 99.91%, versus 88.77% of the optimal NDSI threshold.

Using this approach, Optimized NDVI-NDFSI decision rules for the left six forest land cover types are all determined and listed in Table 2. Together with varying NDFSI thresholds in the table are the OAs of optimized NDVI-NDFSI decision rules and optimal NDSI thresholds. We can see compared with optimal NDSI thresholds, optimized NDVI-NDFSI decision

rules increase OAs greatly. Using the optimal NDSI thresholds, most OAs are ≤ 90%, whereas using the optimized NDVI-NDFSI decision rules, most OAs are ≥ 96%, in line with the accuracies of non-forest land cover types in Table 1.

Table 2 Optimized NDVI-NDFSI decision-rules for forest land-cover types

| Land-cover types | NDFSI thresholds | New | NDSI |
|---|---|---|---|





| (IGBP) | NDVI [-1,-0.1 ) | NDVI [-0.1,0) | NDVI [0.0,0.1 ) | NDVI [0.1,0.2 ) | NDVI [0.2, 0.3 ) | NDVI [0.3, 0.4 ) | NDVI [0.4, 1 ] | OA[a] (%) | OA[b] (%) |
|---|---|---|---|---|---|---|---|---|---|
| Evergreen Needleleaf | -0.18 | 0.12 | 0.05 | 0.06 | 0.16 | 0.24 | 0.31 | 99.80 | 75.92 |
| Forests[c] | -0.09 | -0.09 | -0.28 | -0.10 | 0.06 | 0.19 | 0.26 | 99.72 | 83.85 |
| Deciduous | 0.08 | 0.08 | -0.11 | -0.03 | 0.02 | 0.14 | 0.22 | 99.91 | 54.65 |
| Needleleaf Forests | 0.24 | 0.24 | -0.24 | -0.08 | -0.07 | 0.07 | 0.23 | 99.62 | 88.11 |
| Deciduous Broadleaf | 0.08 | 0.08 | 0.08 | 0.03 | 0.05 | 0.17 | 0.30 | 99.91 | 88.77 |
| Forests | -0.01 | 0.18 | -0.03 | -0.02 | -0.02 | 0.16 | 0.40 | 99.91 | 93.30 |
| Mixed Forests | 0.21 | 0.18 | 0.06 | 0.01 | 0.06 | 0.15 | 0.28 | 99.73 | 85.15 |
|  | 0.28 | -0.09 | -0.10 | -0.03 | 0.01 | 0.15 | 0.29 | 99.69 | 92.48 |
| Woody Savannas | 0.37 | 0.11 | 0.04 | 0.02 | 0.03 | 0.15 | 0.30 | 99.86 | 87.98 |
|  | 0.08 | -0.01 | -0.05 | -0.05 | -0.05 | 0.12 | 0.35 | 99.82 | 94.68 |
| Savannas | 0.29 | 0.13 | 0.07 | 0.06 | 0.04 | 0.24 | 0.36 | 99.91 | 86.43 |
|  | 0.20 | 0.01 | -0.02 | 0.03 | 0.00 | 0.18 | 0.32 | 99.82 | 88.20 |
| Permanent Wetland | 0.50 | 0.19 | 0.12 | 0.17 | 0.31 | 0.35 | 0.35 | 85.53 | 81.42 |
| Savannas | 0.42 | 0.18 | 0.07 | 0.15 | 0.47 | 0.54 | 0.54 | 96.41 | 86.29 |

New OA[a] refers to the overall accuracies of optimized NDVI-NDFSI decision-rules, and NDSI OA[b] refers to those of optimal NDSI thresholds. At Note c, please note rules over every forest land-cover type are conducted for Terra and Aqua separately: the first low is for Terra and the second low is for Aqua.

### 3.1.4 Postprocessing based on surface temperature and DEM

Ice clouds have similar optical properties to snows. From time to time pixels contaminated by broken ice clouds are mistaken for snow pixels, which will lead to some false snow existences even in the tropics. Like the standard MODIS products (Riggs et al., 2006), we also use the postprocessing based on surface temperature and DEM to restore these snow pixels, but here ERA5 reanalysis LST is used rather than the brightness temperature from band 31. According to our previous investigation (Hao et al., 2021), snow pixels will be readjusted as snow-free pixels if their corresponding LST is ≥ 275 K and DEM is ≤ 1300 km, or LST is ≥ 281 K and DEM is ≥ 1300 km.

### 3.2 New Terra-MODIS and Aqua-MODIS SCE datasets under clear skies

Using the aforementioned decision rules, we produce the new clear-sky Terra-MODIS and Aqua-MODIS SCE datasets over China based on the standard MOD09GA and MYD09GA products. Figure 4 gives the flowchart of producing the two datasets, and there are four types of pixels in the datasets, namely snow, snow-free, water body, and gap (mainly induced by cloud pixels).





### 3.3 Cloud-gaps removals

Optical remote sensing is affected by clouds severely, which will result in a large percentage of cloud gaps (often up to 70%) in the preliminary datasets (Huang et al., 2020). Compared with those datasets only under clear skies, a high-quality

continuous SCE dataset undoubtedly has a wider application prospect. In order to generate a complete daily SCE dataset, the following steps are adopted.

### 3.2.1 Aggregation of Terra-MODIS and Aqua-MODIS SCE datasets

There may be different weather conditions when Terra and Aqua pass in one day, and their synergy can reduce cloud gaps approximately by $10 - 30\%$ over China (Huang et al., 2016). Thus, in the study, we will first aggregate the new Terra-

MODIS and Aqua-MODIS SCE datasets to preliminarily exclude some cloud gaps. The aggregating rule is in one day clear information no matter from Terra-MODIS or Aqua-MODIS will be reserved, and the cloud gap is kept still only if cloud existences are indicated by both.

### 3.3.2 Gap-filling based on the HMRF technique

For the aggregated SCE data, we will adopt the method based on the spatio-temporal Hidden Markov Random Field (HMRF)

proposed by Huang et al. (2018) to further fill gaps. The core of this method is computing the total spatio-temporal "energy" (probability) of belonging to snow pixels or snow-free pixels at one specific gap, using

$$U(Snow) = U_{st} (Snow / N_s, N_t) \tag{2}$$

$$U(Snow\text{-}free) = U_{st} (Snow\text{-}free / N_s, N_t) \tag{3}$$

where $U_{st}$ is the spatio-temporal neighbourhood energy function. $N_s$ and $N_t$ denote the spatial neighbourhood and temporal

neighbourhood cantered with the gap, respectively.

Figure 5 illustrates our gap-filling process based on the HMRF technique. For a given gap, we will first calculate the U(Snow) and U(Snow-free) based on the 3 (row) $\times 3$ (column) $\times 3$ (day) spatio-temporal neighbourhood. If the U(Snow) is > the U(Snow-free), the gas will be classified as snow pixels; else it will be classified as snow-free pixels. If there are not sufficient valid pixels for calculating the U(Snow) or U(Snow-free), we will extend the spatio-temporal neighbourhood to 3

(row) $\times 3$ (column) $\times 5$ (day). If there are still not sufficient valid pixels, the spatio-temporal neighbourhood will expand into 5 (row) $\times 5$ (column) $\times 5$ (day). If the above tries all fail, the gap will be kept still.

The gap-filling technique based on the HMRF provides a rigorous analytical framework for integrating spatial and temporal contexts in an optimal manner, which is accurate and high-efficient for MODIS cloud-gaps filling (Huang et al., 2018). Our investigation shows through this step approximately 80% of the gaps in the aggregated data can be reduced.





### 3.3.3 Eliminating residual gaps with auxiliary data

Although most cloud gaps have been filled after the above two processes, there are still ~ 5% gaps left in the current dataset. In this phase, we will use the auxiliary passive microwave snow-depth product provided by Che et al. (2008) to eliminate all remaining gaps. If collocated snow depth is ≥ 2-cm, then the gap will be assigned a snow pixel; else it will be a snow-free pixel (Hao et al., 2019).

### 3.4 New CGF-MODIS SCE dataset

Based on the above steps, we produce the new 2000-2020 daily CGF SCE dataset from our previous Terra-MODIS and Aqua-MODIS SCE datasets. Figure 6 gives the diagram of producing the new daily CGF SCE dataset. The final cloud-free dataset includes three types of pixels, namely, snow, snow-free, and water body.

The new Terra-MODIS, Aqua-MODIS, and CGF-MODIS SCE datasets together constitute the NIEER MODIS SCE product (Hao et al., 2021). It has a 500-m spatial resolution, a daily temporal resolution, and an instance of snow coverage revealed by it is presented in Figure 7.

## 4 Validation

### 4.1 Accuracy metrics

A confusion matrix similar to Table 3 is used to validate or assess all associated MODIS SCE data in the paper, in which the overall accuracy (OA) represents the percentage that snow and snow-free pixels are correctly classified, the producer's accuracy (PA, corresponding to the omission error, OE) represents the percentage that real snow pixels are classified as such, the user's accuracy (UA, corresponding to the commission error, CE) represents the percentage that pixels classified as snow are also snow in reality. Because the Kappa coefficient have caused too much controversy recently (Pontius and Millones, 2011; Stehman and Foody, 2019), we no longer use it as an accuracy metric here. Meanwhile, to more directly show whether SCE is overestimated or underestimated and to what extent, a metric termed as "bias" is introduced. The bias, defined as the ratio of pixels classified as snow to real snow pixels (Hüsler et al., 2012; Wu et al., 2021), is a relative measure to detect overestimation (values > 1) or underestimation of snow (values < 1).

Table 3 Description of the confusion matrix used to validate or assess the associated SCE products (*SS*, *SN*, *NS*, and *NN* are the numbers in each case)

|  |  | MODIS SCE products | |
|---|---|---|---|
|  |  | Snow | Non-snow |
| Reference | Snow | *SS* | *SN* |
| data | Non-snow | *NS* | *NN* |





$$OA\ (\%) = \frac{SS + NN}{T} \times 100, \text{ where } T = SS + SN + NS + NN$$

$$PA\ (\%) = \frac{SS}{SS + SN} \times 100\,;\ OE\ (\%) = 100 - PA\ (\%)$$

$$UA\ (\%) = \frac{SS}{SS + NS} \times 100\,;\ CE\ (\%) = 100 - UA\ (\%)$$

$$Bias = \frac{SS + NS}{SS + SN} = \frac{PA(\%)}{UA(\%)}$$

Note that nominally 20-year and 362 stations' snow-depth measurements will be used as reference data to validate our product, but in fact validations are conducted only during the snow seasons at one station when the snow-cover days are $\geq 20$, because otherwise, some accuracy metrics would become artificially high due to too many non-snow pixels in a mid-latitude region like China (Metsämäki, 2005).

## 4.2 Overview of the new product's accuracies

In Table 4, an overview is given of validation results for the new MODIS SCE product. We can see on a whole the two new clear-sky datasets are very accurate. For the Terra-MODIS SCE dataset, the OA, OE, and CE are 95.51%, 5.56% and 8.26%, respectively; and for the Aqua-MODIS SCE dataset, the OA, OE, and CE are 94.73%, 8.47%, and 6.78%, respectively. The former has a lower omission error, while the latter has a lower commission error. From the "bias" point of view, the Terra-MODIS dataset slightly overestimates snow cover, and the Aqua-MODIS dataset slightly underestimates snow cover. But 255 from the specific values (1.03 and 0.98), the overestimation and underestimation are marginal. Comparatively speaking, the Aqua-MODIS dataset is inferior to the Terra-MODIS dataset. This is not surprising considering the problem of the Aqua-MODIS band 6 and the fact that CMA snow-depth measurement is only conducted in the mornings.

Although the CGF-MODIS SCE dataset is slightly worse than the two clear-sky datasets, there is not significant difference no matter from any accuracy metric. Here the OA, OE, and CE are 93.15%, 8.25% and 9.83%, respectively. Compared with 260 the metrics shown above, the accuracy drops are basically within $1-2$ percentage points. Moreover, the bias of 1.02 also demonstrate an inconspicuous overestimation. Therefore, our cloud-gaps removing processes are effective and the accuracy may be sustained at a consistent level as the clear-sky datasets.

Table 4 Validation results of the new NIEER MODIS SCE product with reference to CMA ground measurements

|  |  | Terra-MODIS SCE dataset | | Aqua-MODIS SCE dataset | | CGF-MODIS SCE dataset | |
| --- | --- | --- | --- | --- | --- | --- | --- |
|  |  | Snow | Non-snow | Snow | Non-snow | Snow | Non-snow |
| CMA ground measurements | Snow | 139664 | 8227 | 134324 | 12427 | 244005 | 21943 |
|  | Non-snow | 12571 | 302759 | 9769 | 264397 | 26597 | 416366 |
| OA (%) |  | 95.51 | | 94.73 | | 93.15 | |





| | | | |
|---|---|---|---|
| PA (%)/OE (%) | 94.44/5.56 | 91.53/8.47 | 91.75/8.25 |
| UA (%)/CE (%) | 91.74/8.26 | 93.22/6.78 | 90.17/9.83 |
| Bias | 1.03 | 0.98 | 1.02 |

### 4.3 The stability in time

To assess the stability of the NIEER MODIS SCE product in time, the accuracy metrics in every snow season since 2000 are calculated separately. Figure 8 presents the inter-annual fluctuation of these accuracy metrics. From the figure, we can see all of the three datasets have a similar fluctuation characteristic. They always perform a consistent high accuracy in one snow season, and a coincident low accuracy in another snow season. This may be attributed to different snow/non-snow number distributions in nature among different years, and varying sample numbers caused by different ground measurements

available in different years.

The highest accuracy happens in the snow season of 2012, where the OEs of the three datasets are 2.84%, 5.76%, and 4.60%, and the CEs are 5.02%, 3.72%, and 6.01%. Meanwhile, the worse accuracy happens in the snow season of 2018, where the OEs of the three datasets are 9.49%, 12.91%, and 14.95%, and the CEs are 15.42%, 14.96%, and 17.15%. But except this snow season, the OEs and CEs in other seasons are all less than 15%. Relative to OE and CE (namely PA and UA), OA's

fluctuations of the three datasets are smaller. The Terra-MODIS dataset fluctuates within 93% − 97%, the Aqua-MODIS dataset fluctuates within 93% − 96%, and the CGF-MODIS dataset fluctuates within 92% − 95%.

In addition, from the "bias" point of view, the values ranging from 0.94 to 1.07 demonstrate there are neither substantial overestimations nor substantial underestimations in all datasets and snow seasons. Therefore, on average the new product is stable in time and also promising to better serve the climatic, hydrological, and other related studies in China.

### 4.3 The reliability in space

To clarify the new product's reliability in space, this section calculates accuracies specific to each CMA station. With the Terra-MODIS SCE dataset as an example, Figure 9 presents the detailed accuracies at all stations. From the figure, we find, in spite of a very high accuracy seen at most stations, the errors may be relatively larger in the southwest of Northeast China and the north-eastern edges of Qinghai-Tibet Plateau. This can be attributed to their patchy snow-cover features and rugged

terrains (Xiao et al., 2020), where Loess Plateau and Qinghai-Tibet Plateau rapidly transit toward Sichuan Basin and Northeast Plain, respectively. The Aqua-MODIS SCE dataset performs a very similar accuracy pattern like Figure 9, but slightly worse.

Figure 10 further details the accuracies of the CGF-MODIS SCE dataset at all stations. From the figure, we see that in North Xingjiang and the north of Northeast China where the snowpack is stable, the accuracy is high, whereas in the northeast of

Inner Mongolia, the northwest of North China, and the Qinghai-Tibet Plateau where snow may melt rapidly even in winter,





the accuracy is relatively lower. Besides the propagated errors from the two original clear-sky SCE datasets, it is supposed that an unstable snowpack in warmer areas would result in another large uncertainty.

## 5 Discussion

### 5.1 Comparisons with standard MODIS products

Using the same ground reference data, we also evaluate the standard SCE data derived from the newest C6.1 MOD10A1, MYD10A1, MOD10A1F, and MYD10A1F products (see Table 5). Comparing the accuracy metrics listed here and those in Table 4, we can see that these products' accuracies are significantly lower than our product's. If ground measurements are seen as "true" values, the MOD10A1 product's OE and CE are 7.22% and 12.97%, respectively, versus 5.56% and 8.26% of our Terra-MODIS dataset; the MYD10A1 product's OE and CE are 13.78% and 23.78%, respectively, versus 8.47% and

6.78% of our Aqua-MODIS dataset. Our improvement is particularly significant for Terra-MODIS SCE, where the CE is improved by over 17 percentage points. Besides, from the "bias" point of view, there are appreciable overestimations to snow cover in the standard MOD10A1 and MYD10A1 products.

Note that here factual improvement may be much more obvious than shown by the accuracy metric of OA because there are a large number of non-snow pixels (far beyond snow pixels) in a mid-latitude region like China, which would pull up it

dramatically. Although the new product's improvement in OA is within 1% relative to the MOD10A1 products, in fact, the OE has decreased by 37% and the CE has decreased by 22%.

Table 5 Accuracies of SCE products derived from MOD10A1, MYD10A1, MOD10A1F, and MYD10A1F using the same ground measurements

| Standard products | OA (%) | PA (%)/OE (%) | UA (%)/CE (%) | Bias |
| --- | --- | --- | --- | --- |
| MOD10A1 | 95.47 | 92.78/7.22 | 87.03/12.97 | 1.07 |
| MYD10A1 | 93.82 | 86.22/13.78 | 76.22/23.78 | 1.13 |
| MOD10A1F | 89.54 | 88.96/11.04 | 86.74/13.26 | 1.03 |
| MYD10A1F | 84.36 | 78.93/21.07 | 85.53/14.47 | 0.92 |

Due to possible wider application, the CGF products' comparison is emphasized here. Relative to the better standard product

— MOD10A1F, our product's OA increases by nearly 4 percentage points, the OE drops from 11.04% to 8.43%, and the CE drops from 13.26% to 9.83%; relative to the worse standard product — MYD10A1F, the OA even increases by nearly 9 percentage points, the OE drops from 21.07% to 8.25%, and the CE drops from 14.47% to 9.83%. It is clear for the standard CGF products all of the OEs and CEs are larger than 10%, while for our product the OE and CE are both within 10%. Meanwhile, the two standard CGF products' biases are also further away from 1. Therefore, our improvement to CGF SCE

is obvious, too. Besides, comparing the two standard clear-sky products with the two standard CGF products, we find their cloud-gap removing strategy indeed lowers the accuracies dramatically.





## 5.2 Improvement in forest areas

To figure out the improvement in forest areas, the validations and comparisons at four forest CMA stations are isolated and listed in Table 6. From the table, one can see the OA increases by 3 – 10 percentage points, the OE drops by 1 – 8 percentage points, and the CE drops by 4 – 21 percentage points. As well, more significant improvements are seen for the Aqua-MODIS and the CGF SCE. For examples, the CE of the new Aqua-MODIS SCE dataset has already dropped to 9.62%, comparing with 30.13% of the MYD10A1 product; the OA of the new CGF-MODIS dataset increases to 91.23%, compared with 81.79% of the MOD10A1F product. Thus, the improvement of the new product in forest areas is also considerably significant, even more significant than in non-forest areas.

Table 6 Validation and comparison at four forest CMA stations

| SCE products | OA (%) | PA (%)/OE (%) | UA (%)/CE (%) | Bias |
|---|---|---|---|---|
| Terra-MODIS SCE dataset | 95.19 | 97.84/2.16 | 90.74/9.26 | 1.08 |
| MOD10A1 | 92.48 | 96.62/3.38 | 86.01/13.99 | 1.12 |
| Aqua-MODIS SCE dataset | 94.18 | 97.77/2.23 | 90.38/9.62 | 1.08 |
| MYD10A1 | 90.83 | 92.60/7.40 | 69.87/30.13 | 1.33 |
| CGF-MODIS SCE dataset | 91.23 | 93.39/6.61 | 92.12/7.88 | 1.02 |
| MOD10A1F | 81.79 | 85.95/14.05 | 84.10/15.90 | 1.02 |
| MYD10A1F | 82.78 | 89.53/10.47 | 83.12/16.88 | 1.07 |

Two more intuitional examples that demonstrate the improvement of CGF SCE data are shown in Figure 11. One is a typical forest area in Northeast China on Jan. 1st, 2018; another comes from southeast forest regions of Qinghai-Tibet Plateau on Feb. 1st, 2018. It is clear that the new MODIS-CGF SCE dataset agrees much better with the higher-resolution snow maps than the standard CGF product does. The standard MOD10A1F perceivably overestimates snow coverage in these forest areas.

Of course, the new product is far from perfect. In particular, the product's performance is limited severely by the cloud mask provided by the standard MOD09GA/MYD09GA products, which is virtually caused by the cloud/snow confusion problem. On the one hand, some broken ice clouds are susceptible to being mistaken for snow pixels because of their similar optical properties, which will result in some artificial snow pixels in South China (although it has been mitigated largely compared with the standard MODIS snow products). On the other hand, snow pixels are also possibly mistaken for clouds, which will result in some omitted snow covers. During our validations or comparisons, we found this phenomenon is somewhat common in the edges of snow-cover areas and the forest areas of Northeast China.

## 6 Summary and outlook

Under the support of several national programs of China, we have produced a new MODIS SCE product over China, which is committed to addressing currently known problems of the standard snow products.



Toward this, the optimal NDSI thresholds varying with land cover types are extracted, and the NDVI-NDFSI decision rules proposed by Wang et al. (2015) specific for snow discrimination in forest areas are optimized, both by a number of training samples indicated by the higher-resolution snow maps from Landsat-8 OLI images. Meanwhile, to obtain a daily continuous cloud-free SCE dataset, the clear-sky snow SCE data from Terra-MODIS and Aqua-MODIS are aggregated first; an HMRF gap-filling technique, which can simultaneously assimilate temporally and spatially neighbouring information, is imported second; and finally, a totally cloud-free SCE is mapped through replacing the residual gaps with auxiliary passive microwave snow-depth data.

The new product is named the NIEER MODIS SCE product, which contains three individual datasets — the Terra-MODIS SCE dataset, the Aqua-MODIS SCE dataset, and the MODIS CGF SCE dataset. The first two provide the SCE knowledge under clear skies directly derived from the MODIS surface reflectance products, while the last provides daily and totally continuous SCE knowledge by a series of processes filling cloud-induced gaps.

The comprehensive validations against 362 CMA stations across China have revealed that our product is not only very accurate but also considerably stable in time and reliability in space. On a whole, there are neither any significant overestimations nor any significant underestimations in all datasets, and the cloud-gaps removing processes do not lower the SCE accuracy clearly. Compared with SCE indicated by the standard MODIS snow products, our product's accuracies are obviously higher, especially for the Aqua-MODIS and CGF SCE. Relative to the standard MYD10A1 product, the CE drops 17 percentage points, from 23.78% to 6.78%. Relative to the better MOD10A1F CGF product, the OA increases by nearly 4 percentage points, from 89.54% to 93.15%. Meanwhile, as we expected the improvement in forest areas is also evident. If validations at forest stations are isolated, OA increases by 3 − 15 percentage points. Therefore, the new product has provided more reliable knowledge on snow coverage over China, and thereby will be a better choice for climatic, hydrological, and other related studies there.

The problem of cloud/snow confusion may contribute to the largest uncertainty in the new product. There are many cases that snow pixels are mistaken for cloudy pixels or the opposite cases due to the inaccurate cloud mask provided by the MOD09GA/MYD09GA products. In the future, we will consider further improving our product from this aspect.

**Code and data availability**

The new MODIS SCE product is available from the National Tibetan Plateau Data Center here: https://dx.doi.org/10.11888/Snow.tpdc.271387 (Hao et al., 2021). The code used to generate the product can be obtained from the authors without conditions.

**Author contribution**

XH, GH, and ZZ conceived and designed the study. XS, WJ and HZ developed the code and generated the product on GEE platform. XH and GH prepared the initial draft of the manuscript. All authors contributed to the data analysis, results' interpretation and validation, and revision of the manuscript.

**Competing interests**

The authors declare that they have no conflict of interest.



**Acknowledgements**

We would like to thank the China Meteorological Administration (CMA) for their reliable ground measurements, and the Google Earth Engine (GEE) for their high-quality services.

**Financial support**

This work was jointly supported by the National Key Research and Development Program of China (Grant No. 2019YFC1510503), the National Natural Science Foundation of China (Grant No. 41971325; 42171322; 42171391), and the

Science & Technology Basic Resources Investigation Program of China (Grant No. 2017FY100502).

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

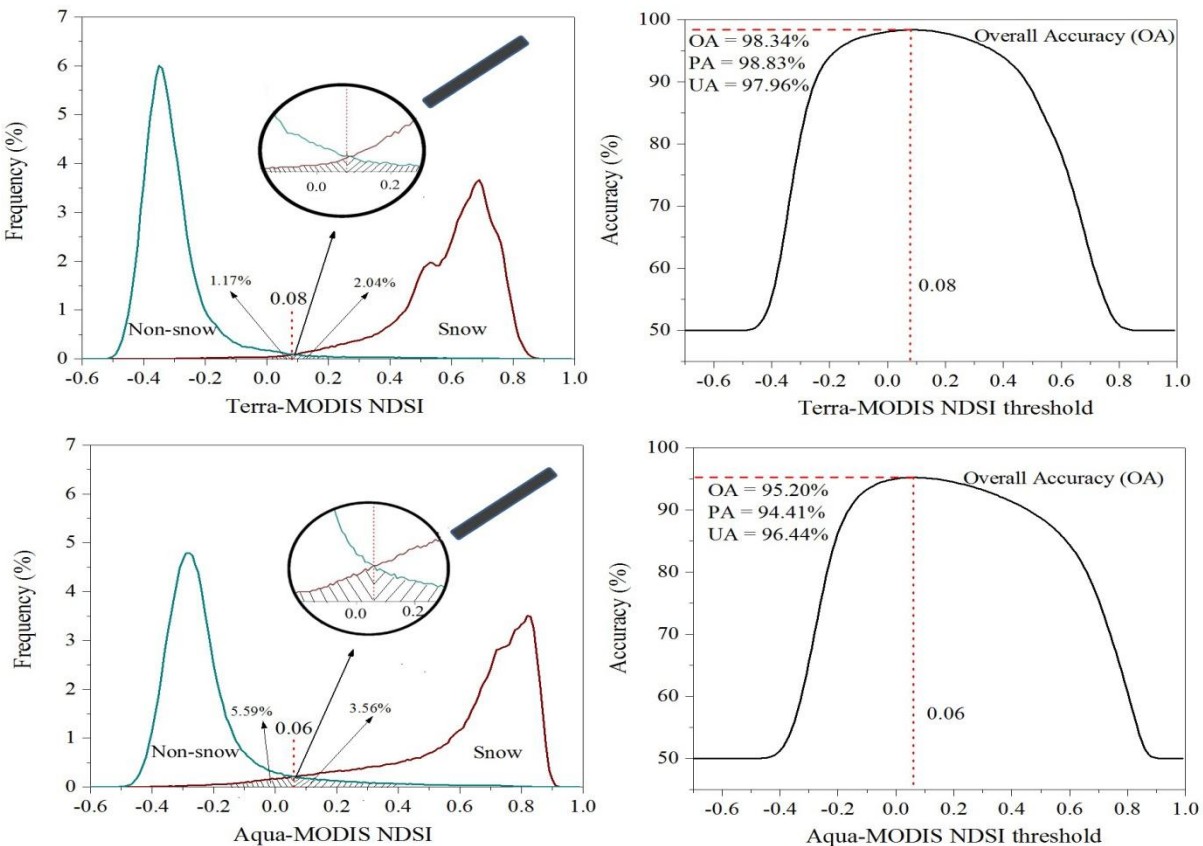





**Figure 1: Optimal NDSI thresholds over the land cover type of "Barren or Sparsely Vegetated" (The left is NDSI frequency distribution, and the right is the fluctuation of overall accuracy as a function of NDSI threshold.)**

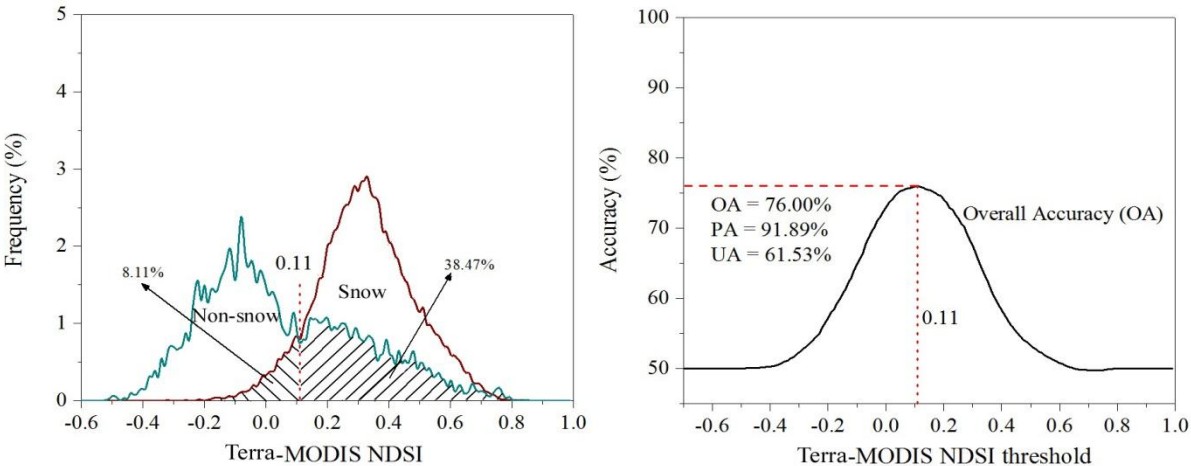

**Figure 2: Same as Figure 1 but over the land cover type of "Evergreen Needleleaf Forest"**

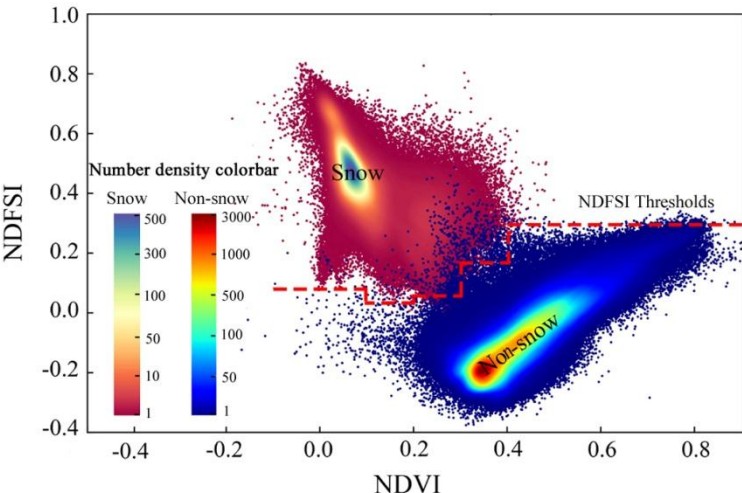

**Figure 3: NDVI-NDFSI number-density scatterplot over "Deciduous Broadleaf Forest" from Terra-MODIS training samples (Colours represent the number-density within a 0.01×0.01 bin)**





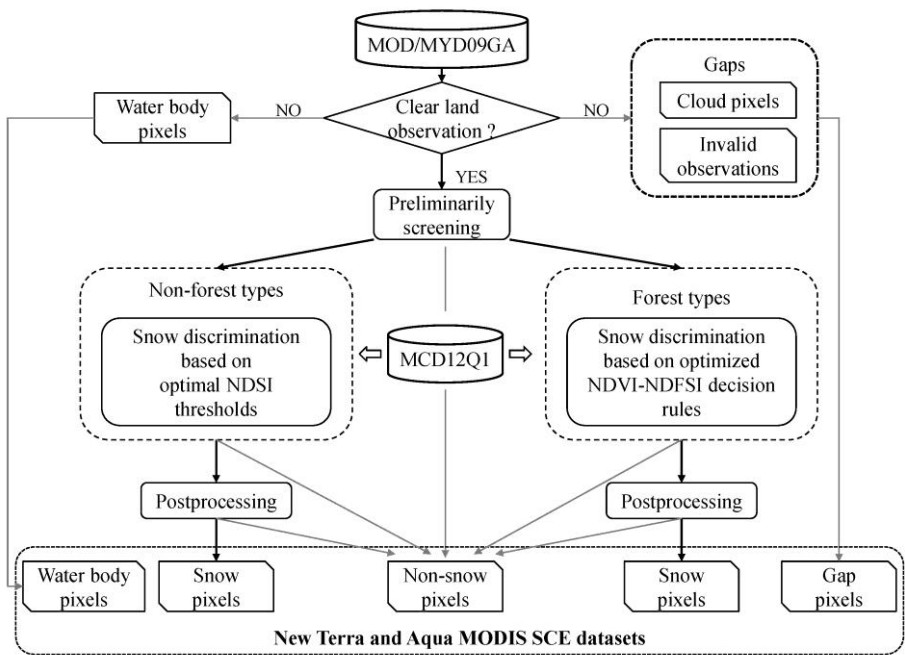

**Figure 4: Flowchart of producing the new Terra-MODIS and Aqua-MODIS SCE datasets**

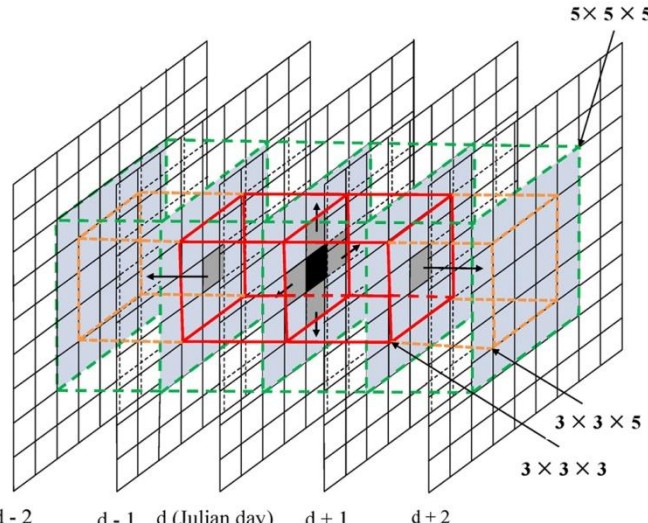

**Figure 5: Diagram of the HMRF gap-filling process used in the study (Dark-grey represents high weight, and light-grey represents low weight)**





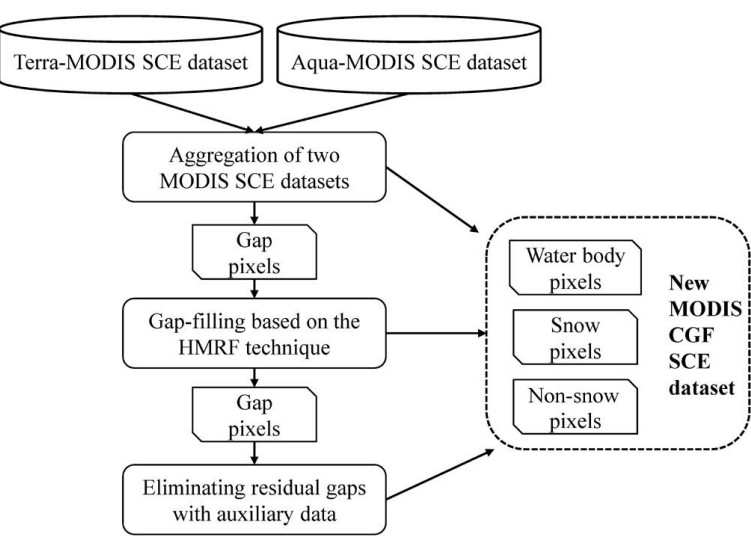

**Figure 6: Schematic of producing the new MODIS CGF SCE datasets**

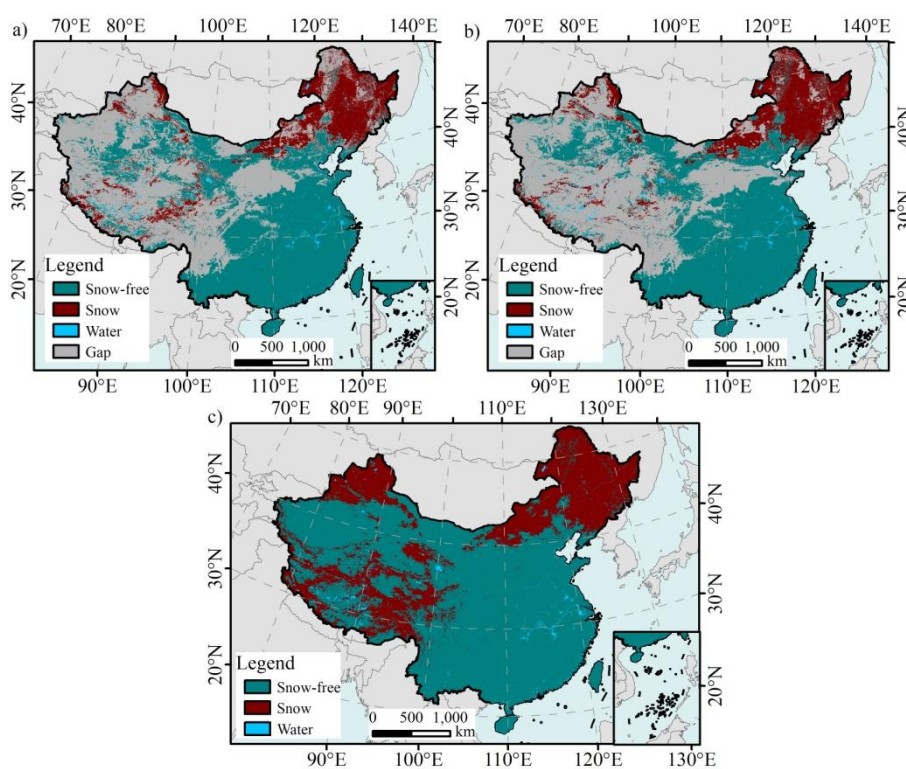

**Figure 7: NIEER MODIS SCE product on Jan. 4, 2020: a) Terra-MODIS SCE dataset; b) Aqua-MODIS SCE dataset; c) CGF-MODIS SCE dataset**





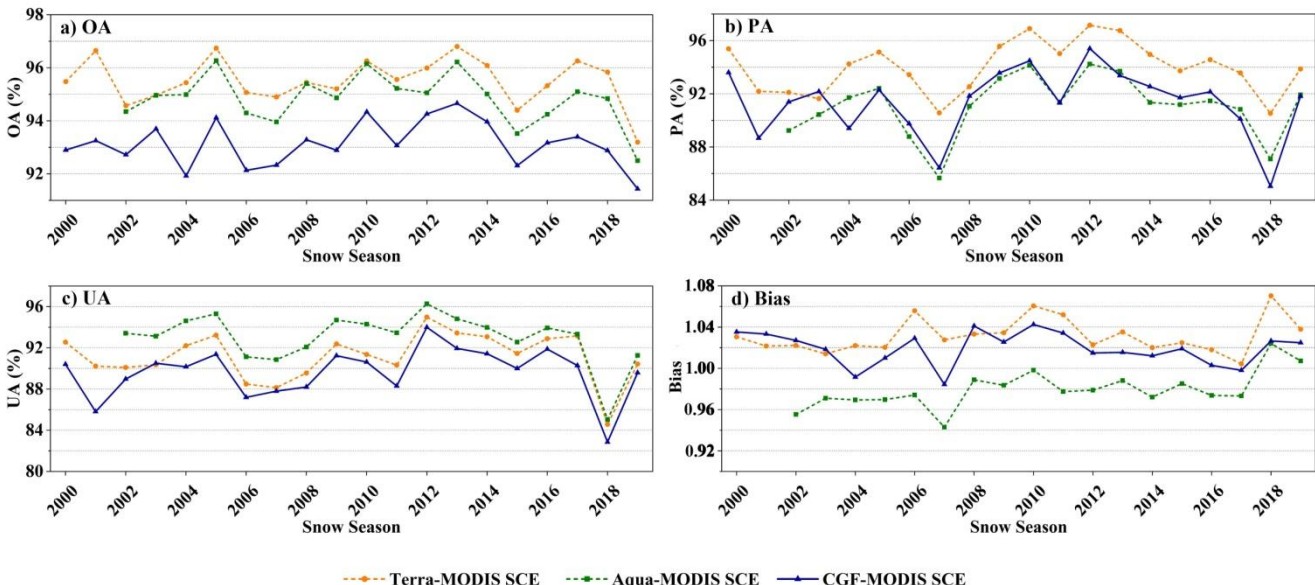

**Figure 8: Annual accuracy metrics' fluctuation of the NIEER MODIS SCE product: a) OA; b) PA; c) UA; d) Bias**



Figure 9: Accuracies of the Terra-MODIS SCE dataset at each CMA station: a) OA; b) PA; c) UA; d) Bias





**Figure 10: Accuracies of the CGF-MODIS SCE dataset at each CMA station: a) OA; b) PA; c) UA; d) Bias**



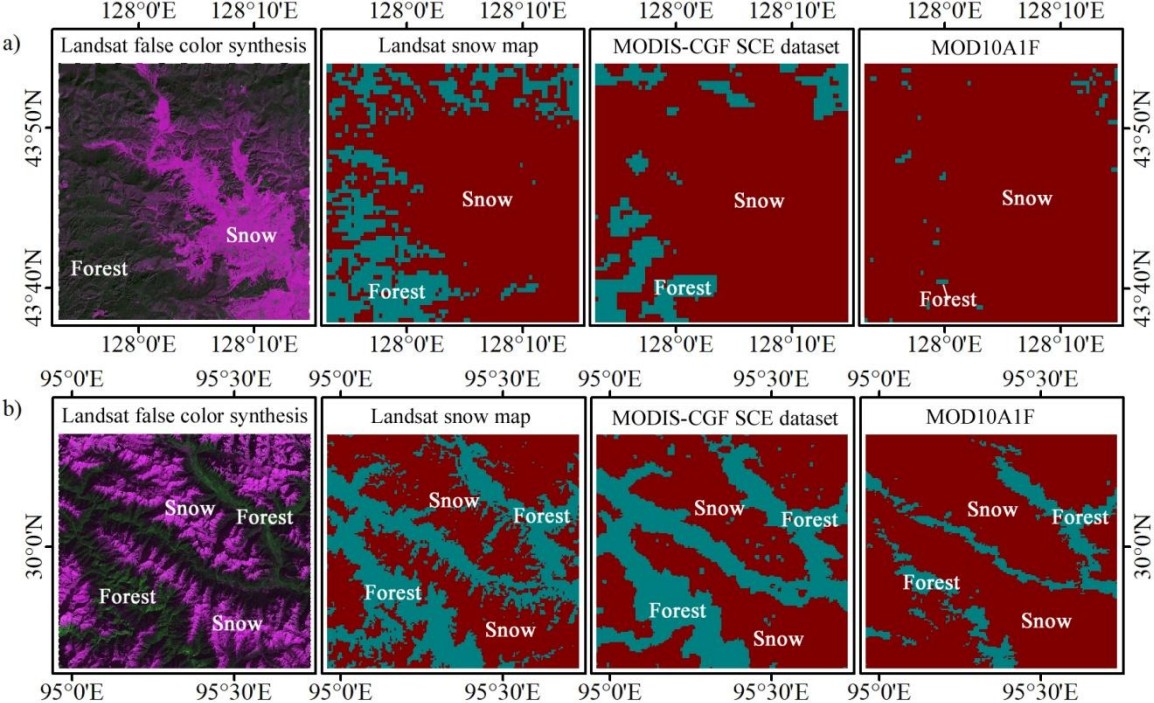

**Figure 11: Intuitional improvements of CGF SCE in two representative forest areas: a) an example in Northeast China forest region; b) an example in Qinghai-Tibet Plateau forest region**
