# Peer review of "Development and validation of a new MODIS snow-cover-extent product over China"

_Hydrology and Earth System Sciences, 2021_

## Referee Comment (RC1)

**Overall comments**

Hao's paper produced a daily cloud-free snow cover extent product with 500 m spatial resolution based on MOD09GA and MYD09GA land surface reflectance dataset in China. The new daily cloud-free snow product was also verified by snow depth observation from climate stations. The results show that the new snow cover product with higher overall accuracy than the standard MODIS daily snow product released by NSIDC, which provides a more reliable snow dataset for other related studies.

So far as we know, the NDSI threshold is the crucial parameter for snow detection by use of optical remote sensing data. However, the NDSI threshold varies with the snow depth, snow fraction, pollution, as well as different land-use types, which cause great difficulty in remote sensing mapping of snow cover at present. From the content of this paper, this paper analyzes the sensitivity of NDSI, that is, the effect of different land cover types on NDSI, then to find the optimal NDSI threshold for different land cover types in the study area. Especially in forest areas, the NDVI and NDFSI were involved in decision tree classification.

The paper is scientifically sounding. I hesitate to say major revisions since it is mostly about reorganizing the paper structure and a few data process issues need to clarify, but quite a few minor revisions should be undertaken, and the editorial changes for language usage throughout the manuscript need to be addressed before publishing this manuscript to journal of Hydrology Earth System Sciences.

**General Comments**

1. The structure of the paper is not following the convention, the results and methods are mixed. I think the paper requires a better structural organization to improve its readability.

2. A large number of Landsat snow data are used in this paper, partly for training and partly for verification. But why is the new MODIS snow product verified eventually by the ground observation data from meteorological stations?

3. According to the characteristic curve of the snow refection spectrum, snow cover usually has a high NDSI value in traditional cognition, but the NDSI value calculated by using surface reflectance in this paper shows that the NDSI of snow cover under

most land cover types is relatively low. If this is true because the MODIS standard product uses zenith reflectance rather than an atmospherically corrected product to calculate NDSI, the authors should highlight this conclusion in the Conclusion section.

**Minor Comments**

1. L15, surface reflectance data, a new daily MODIS snow cover extent (SCE) product from 2000 to 2020 over China has been produced.

2. L21, Against 362 China Meteorological Administration (CMA) stations, the validation results show…

3. L24, Biases ranging from 0.98 to 1.02, indicating that the SCEs given by the new snow product are neither overestimated nor underestimated significantly.

4. L26, clearly – obviously

5. L55,NDSI >=0.4

6. L57, The C6 snow product provides a standardized NDSI but does not redefine a new threshold of snow cover.

7. Suggest simplifying L48-57, and focusing on the idea of cloud removal algorithm in MOD10A1F snow product.

8. Line58,While the released MODIS snow products have…

9. L64-66,need a reference here.

10. Line68, add a summary of the current cloud removal algorithms for MODIS daily snow product.

11. L88-89, which version and period of MOD09GA and MYD09GA were used, and which year of MCD12Q1 was used in this study.

12. L90, land surface reflectance

13. L94, how do the Landsat OLI snow maps come from? Suggest making a spatial distribution map of Landsat images, distinguishing training data sets from validation data sets, and also including meteorological stations.

14. L107, resampled, and delete 'or aggregated'

15. L109, Provide the period of climate stations data used.

16. Change the headline of the 3 sections to 'Method'.

17. L133-135. Please statement as clear as possible.

18. L136-138, during the preliminarily screening, what is the internal purpose of each threshold? Please explain.

19. In 3.1.2, put Table 1 into the results section, this part only focuses on the method used in this study. And the title of the table is 'Optimal NDSI thresholds over eight non-forest land-cover types', but the content of the table has the forest land cover, such as 'Evergreen Broadleaf Forest'.

20. In 3.1.3, the same suggestion is with 3.1.2.

21. L162, Eq. (1) is about NDSI, not NDFSI.

22. In 3.1.4, can you add a reference for the accurate evaluation of EAR surface temperature products? The LST and elevation thresholds were used for snow misclassification in high elevation areas in MOD/MYD10A1 V006, not for ice clouds.

23. L190, change the 'gap' to 'data gap'. And please clarify the source of the data gap, such as the cloud.

24. In 3.3.2, the results of Figure 5 should be in the Results section, here only focus method.

25. Combine the 3.2 and 3.4 into one section, and suggest deleting the content of the 3.2, 3.3, and 3.4 section.

26. Change the headline of 4 to Results.

27. Section '4.1 Accuracy metrics' should be in the Method section.

28. L258-262. Dose the Terre/Aqua MODIS SEC dataset were validate by ground measurements under clear sky?

29. In the 4.3 section, the accuracy varies from year to year, mainly due to ground observation, as the authors mentioned. Any other explanations?

---

## Author Comment (AC1)

**Overall comments**

Hao's paper produced a daily cloud-free snow cover extent product with 500 m spatial resolution based on MOD09GA and MYD09GA land surface reflectance dataset in China. The new daily cloud-free snow product was also verified by snow depth observation from climate stations. The results show that the new snow cover product with higher overall accuracy than the standard MODIS daily snow product released by NSIDC, which provides a more reliable snow dataset for other related studies.

So far as we know, the NDSI threshold is the crucial parameter for snow detection by use of optical remote sensing data. However, the NDSI threshold varies with the snow depth, snow fraction, pollution, as well as different land-use types, which cause great difficulty in remote sensing mapping of snow cover at present. From the content of this paper, this paper analyzes the sensitivity of NDSI, that is, the effect of different land cover types on NDSI, then to find the optimal NDSI threshold for different land cover types in the study area. Especially in forest areas, the NDVI and NDFSI were involved in decision tree classification.

The paper is scientifically sounding. I hesitate to say major revisions since it is mostly about reorganizing the paper structure and a few data process issues need to clarify, but quite a few minor revisions should be undertaken, and the editorial changes for language usage throughout the manuscript need to be addressed before publishing this manuscript to journal of Hydrology Earth System Sciences.

Response: Thank you very much for the time and effort put into reviewing our manuscript. We will try our best to make the manuscript better. Your comments really help us a lot.

We know as a non-native speaker there must be language problems throughout the manuscript, more or less. Therefore, before the submission to HESS it in fact had been edited by a professional company, EDITSPRINGS. Please see the below certificate (at that time, our manuscript title is "The NIEER MODIS snow extent product over China"). After this revision, we asked them to polish the English again.

[Figure]

**EDITORIAL CERTIFICATE**

This document certifies that the manuscript listed below was edited for proper English language, grammar, punctuation, spelling, and overall style by one or more of the highly qualified native English speaking editors at EditSprings.

Manuscript title:

The NIEER MODIS snow cover extent product over China

Authors:

Hao Xiaohua, Huang Guanghui, Zheng Zhaojun, Sun Xingliang, Ji Wenzheng,
Zhao Hongyu, Wang Jian, Li Hongyi, Wang Xiaoyan

Date Issued:

Aug 27 2021

Certificate Number:

ES-202106241645578090

This certificate can be verified on www.editsprings.com/query.asp. This document certifies that the manuscript listed above was edited for proper English language, grammar, punctuation, spelling, and overall style by one or more of the highly qualified native English speaking editors at EditSprings. Neither the research content nor the authors' intentions were altered in any way during the editing process. Documents receiving this certification should be English-ready for publication; however, the author has the ability to accept or reject our suggestions and changes.

**General Comments**

1. The structure of the paper is not following the convention, the results and methods are mixed. I think the paper requires a better structural organization to improve its readability.

Response: Very good suggestion. We have reorganized the paper according to your comments. In section 3, we only preset the algorithm we have developed in the paper, but introduce the new product in section 4 simply. Because one of our most important objectives is producing a new SCE product, we indeed need a separate section to show these contents.

2. A large number of Landsat snow data are used in this paper, partly for training and partly for verification. But why is the new MODIS snow product verified eventually by the ground observation data from meteorological stations?

Response: Our previous arrangements around these contents may mislead you. We have reorganized them. Please see the new sections 2.2 and 3.1. We use the Landsat OLI snow maps only to obtain training samples and subsequently refine the snow detection rules, rather than partly for training and partly for verification. This is done for the following reasons. First, it seems unreasonable to use the data from the same source for both training and validating, and one may certainly think your validation is better because the same source training data are utilized. Second,

Landsat OLI snow maps are obtained usually in clear skies, which indicates the comparisons or validations are conducted only in clear skies. This will result in a very similar accuracy no matter for clear products and cloudy products and may bring a larger confusion in the section of validation. After all, the CGF SCE is our final object. Isn't it sounder, using Landsat OLI snow maps to obtain training samples but using the stations' observations to validate the final products?

3. According to the characteristic curve of the snow refection spectrum, snow cover usually has a high NDSI value in traditional cognition, but the NDSI value calculated by using surface reflectance in this paper shows that the NDSI of snow cover under most land cover types is relatively low. If this is true because the MODIS standard product uses zenith reflectance rather than an atmospherically corrected product to calculate NDSI, the authors should highlight this conclusion in the Conclusion section.

Response: Yes, for a totally snow covering pixel generally its NDSI is very high, but for a mixed pixel that fractional snow cover is > 0.5, the NDSI may be not that high. In our study, we find fixed pixels are very common. We think this may be caused by the following two reasons. First, relative to small TM or ETM pixels, spatially much larger MODIS pixels undoubtedly are even inclined to be mixed. Second, over a mid-latitude region like China the snow distribution is always patchy, and such a snow cover feature also results in more mixed pixels. Due to the problem of mixed pixels, the optimal NDSI threshold may be pulled down dramatically, much less than 0.4 that previously is thought as best. Lower NDSI thresholds are not unusual in literature. For example, the study of Zhang et al (2019) concluded an optimal NDSI threshold of 0.1, which is very close to our thresholds shown in table 1. We don't think the discrepancy of TOA and surface NDSIs will influence the threshold determination significantly except in some extreme cases (such as heavy aerosols).

Zhang, H. B., Zhang, F., Zhang, G. Q., Che, T., Yan, W., Ye, M., and Ma, N.: Ground-based evaluation of MODIS snow cover product V6 across China: Implications for the selection of NDSI threshold, Sci. Total Environ., 651, 2712-2726, doi:10.1016/j.scitotenv.2018.10.128, 2019.

**Minor Comments**

1. L15, surface reflectance data, a new daily MODIS snow cover extent (SCE) product from 2000 to 2020 over China has been produced.

Response: Thanks. We have modified the sentence accordingly. But we think the word "daily" may be inappropriate used here because strictly only the final CGF dataset is daily and the first two datasets are not.

2. L21, Against 362 China Meteorological Administration (CMA) stations, the validation results show…

Response: Very good suggestion. We have modified the sentence accordingly.

3. L24, Biases ranging from 0.98 to 1.02, indicating that the SCEs given by the new snow product are neither overestimated nor underestimated significantly.

Response: Thanks. We have modified the sentence accordingly.

4. L26, clearly – obviously

Response: Done. Thanks.

5. L55,NDSI >=0.4

Response: Very good suggestion. We have modified the sentence accordingly.

6. L57, The C6 snow product provides a standardized NDSI but does not redefine a new threshold of snow cover.

Response: Yes, the C6 snow products do not redefine the new threshold explicitly. But there is a sentence on page 13 in the "C6_MODIS_Snow_User_Guide" saying "Pixels detected with snow cover in the 0.0 < NDSI < 0.10 are reversed to a 'not snow' result and bit 2 of the NDSI_Snow_Cover_Algorithm_Flags_QA is set. That bit flag can be used to find where a snow

cover detection was reversed to 'not snow.'" From this sentence and the User Guide, 0.1 in fact is indicated.

Riggs, G.A., Hall, D.K., Roman, M.O.: MODIS snow products user guide for collection 6, http://modis-snow-ice.gsfc.nasa.gov/?c=userguide, 2016.

7. Suggest simplifying L48-57, and focusing on the idea of cloud removal algorithm in

MOD10A1F snow product.

Response: We reorganized these paragraphs and sentences. Our arrangements are: first, introducing the standard MOD10A1F and MYD10A1F under clear skies, and then pointing out their shortcomings; second, introducing the standard CGF products, and then analyzing their shortcoming; and finally in the last paragraph of the section "introduction" presenting what we will do to improve or mitigate these shortcomings. Therefore, the idea of cloud removal algorithm adopted by MOD10A1F is shown in the next paragraph.

8. Line58,While the released MODIS snow products have…

Response: Thanks. We have deleted this sentence.

9. L64-66,need a reference here.

Response: Thanks. We have added a such reference.

10. Line68, add a summary of the current cloud removal algorithms for MODIS daily snow

product.

Response: Thanks. We have added a summary sentence at the end of this paragraph.

11. L88-89, which version and period of MOD09GA and MYD09GA were used, and which year

of MCD12Q1 was used in this study.

Response: Thanks. We have added these information. Please see the revised section 2.1.

12. L90, land surface reflectance

Response: Done. Thanks.

13. L94, how do the Landsat OLI snow maps come from? Suggest making a spatial distribution map of Landsat images, distinguishing training data sets from validation data sets, and also including meteorological stations.

Response: We have revised section 2.2 and also rescheduled the introduction on MOD09GA and MYD09GA training samples here. The previous separate arrangements may mislead you. We use the Landsat OLI snow maps only to refine the snow detection rules, rather than to validate our products. Please also see our above response to the general comment 2. In the initial draft, we indeed give such a map to show the spatial distribution of all OLI snow maps. But there are too many scenes of OLI snow maps (1509 for Terra and 1648 for Aqua) and placing them together will make the figure so messy that little information can be seen. Please see the following figure. Isn't it ugly?

[Figure]

Besides, from figure 9 and 10 which show the accuracies of the Terra-MODIS SCE and CGF-MODIS SCE datasets at each CMA station, one can easily see the distribution of these stations. Therefore, this information may seem unnecessary here.

14. L107, resampled, and delete 'or aggregated'

Response: We have changed the "re-sampled" into "resampled", but not deleted "or aggregated"

because for the first two products whose resolution is much larger than 500m the word "resample" is reasonable but for the last product whose resolution is much less than 500m the word "aggregate" is more proper.

15. L109, Provide the period of climate stations data used.

Response: Done, please note the words "since 2000".

16. Change the headline of the 3 sections to 'Method'.

Response: Done. Thanks.

17. L133-135. Please statement as clear as possible.

Response: We have reorganized them. Please see the revised manuscript.

18. L136-138, during the preliminarily screening, what is the internal purpose of each threshold? Please explain.

Response: As we have mentioned in line 130 in the previous manuscript, the purpose of the preliminarily screening is to preclude the pixels that are impossibly covered by snow completely. These thresholds are combined together to achieve this goal. This is a step also adopted by the MODIS standard products. The internal implication is that snow is impossible to own too low visible reflectances and too high band 6 reflectance.

19. In 3.1.2, put Table 1 into the results section, this part only focuses on the method used in this study. And the title of the table is 'Optimal NDSI thresholds over eight non-forest land-cover types', but the content of the table has the forest land cover, such as 'Evergreen Broadleaf Forest'.

Response: Here it is a very simple decision rule, namely, pixels whose NDSI value is ≥ the NDSI threshold will be identified as snow cover, and otherwise they are snow-free. Therefore, in this section the optimal NSDI threshold is crucial. We'd better directly present the results here, we

think. Yes, in table 1 there is an exception, "Evergreen Broadleaf Forest". We have explained it in the text. This may be resulted by its sparse number in China.

20. In 3.1.3, the same suggestion is with 3.1.2.

Response: Thanks. But please see the above response.

21. L162, Eq. (1) is about NDSI, not NDFSI.

Response: Yes, but we have pointed that NDSFI is using band 2 to substitute band 4 in Eq. (1).

22. In 3.1.4, can you add a reference for the accurate evaluation of EAR surface temperature products? The LST and elevation thresholds were used for snow misclassification in high elevation areas in MOD/MYD10A1 V006, not for ice clouds.

Response: We have added a such reference. Our previous expressions may mislead you. We have modified them. Initially, LST is introduced to screen out the false snow pixels that may be led by ice clouds no matter what its DEM is (C5 snow products). But later they find there may be warm snows existing over highlands. Therefore, DEM is also introduced in C6. Please see the following sentences in the user guide. If snow is detected in a pixel at height < 1300 m and that pixel has an estimated band 31 brightness temperature (BT) ≥ 281 K, that snow detection decision is reversed to 'not snow'. In C6 the surface temperature screen is combined with surface elevation and is used in two ways. This combined screen reverses snow cover detection on low elevation < 1300 m surfaces that are too warm for snow and the algorithm QA bit flag is set. Snow cover detection at ≥1300 m on a surface that is too warm for snow is not reversed but that snow cover detection is flagged as too warm by setting the algorithm QA bit flag.

23. L190, change the 'gap' to 'data gap'. And please clarify the source of the data gap, such as the cloud.

Response: Done. Thanks.

24. In 3.3.2, the results of Figure 5 should be in the Results section, here only focus method.

Response: We have reorganized these sections.

25. Combine the 3.2 and 3.4 into one section, and suggest deleting the content of the 3.2, 3.3, and 3.4 section.

Response: Thanks for your suggestion. We have reorganized the manuscript according to your advice.

26. Change the headline of 4 to Results.

Response: Done. Thanks!

27. Section '4.1 Accuracy metrics' should be in the Method section.

Response: We have reorganized these sections.

28. L258-262. Dose the Terre/Aqua MODIS SEC dataset were validate by ground measurements under clear sky?

Response: Yes. Terre/Aqua MODIS SEC can only provide the snow cover conditions under clear-skies. Therefore, their validations are conducted in clear skies.

29. In the 4.3 section, the accuracy varies from year to year, mainly due to ground observation, as the authors mentioned. Any other explanations?

Response: Besides ground observation, the contrast of snow days versus snow-free days may also impact the accuracy. If snow days are more and snow-free days are less in one year, then in this year the commission will prone to be larger. Otherwise, omission error will be larger. The over accuracy will be better only if the contrast of snow days versus snow-free days reaches one balance.

---

## Author Comment (AC2)

This paper produced a new MODIS snow cover product over China. This product includes Terra SCE and Aqua SCE datasets, as well as a cloud-gap-filled SCE dataset. Validation against with in situ snow depth measurements, these products show obvious improvements than standard MODIS SCE products. The produced snow cover extent product could be a significant dataset for studying climate change over China.

Despite of its significance, several issues still need to be resolved before a possible publication to HESS. 1) OLI snow product is very important for optimal NDSI thresholds, and how do you obtain them? You should describe it more sufficiently; 2) Organization of the paper should be improved, especially section 3. I am confused by the first two paragraphs of section 3. Why you introduce how to obtain the samples here? I think maybe putting them in section 2 is better. More importantly, you should show a flowchart first in section 3.1. Otherwise, it is difficult for readers to understand how you will produce SCE from MOD09GA or MYD09GA. There is a same problem for section 3.2. I would suggest you reorganize this section. 3) Some minor errors that may need to be modified are listed as follows.

Response:Many thanks for your reviewing. We will try our best to modify the manuscript according to your comments and suggestions. About the first suggestion, we have added a sentence to simply introduce how we obtain the OLI snow products. that is "*The first group is derived from 1509 scenes of OLI images, which will be regarded as "true" values to acquire the Terra-MODIS training samples; and the second group comes from 1648 scenes of OLI images, which will be used to acquire the Aqua-MODIS training samples.*

*These snow maps are generated by the improved "SNOMAP" algorithm developed by Chen et al. (2020), and have a spatial resolution of 30-m.*" in the revised manuscript.

About the second suggestion, it is really very good, and we have reorganized the manuscript following your detailed advices. First, we have deleted the contents of how to obtain the samples (first two paragraphs) in section 3, but added them into section 2.2. Please see the new section 2.2, "Landsat-8 OLI snow maps and MOD09GA \ MYD09GA training samples". Second, we reorganized section 3.1 and section 3.2. This time, we first present the flowcharts, and then give the key steps. Please see the blow paragraphs in the revised manuscript.

*Guided by the algorithm of the MODIS standard snow products (Hall et al., 2002; Riggs et al., 2006; Riggs et al., 2016 ) and our motivations that are mentioned in section 1, we develop a new snow discrimination algorithm for clear-skies which is shown in figure 1. Approximately, it contains four steps to finally determine snow-cover conditions from MODIS clear-sky surface reflectance data. The very first step is preliminarily screening with the purpose of precluding the cases that are impossibly covered by snow completely. The second step provisionally determines snow-cover conditions over non-forest land-cover types using the optimized NDSI thresholds; while the third step determines snow-cover conditions over forest land-cover types through importing a new decision rule. Step four is postprocessing based on surface temperature and DEM, which is designed to reverse those false snow pixels determined by the previous two steps into snow-free pixels. Among these steps, step two and three are crucial for snow discrimination under clear skies, and will be emphasized in the paper.*

*Figure 5 describes the flow of the new cloud-gap removing algorithm we developed in the study. It*

*can be divided into three steps. The first step is preliminarily excluding some cloud gaps by the synergy of Terra-MODIS and Aqua-MODIS; the second step is further filling gaps according to the implication of the nearby clear-sky pixels using Hidden Markov Random Field (HMRF) technique; and in step three all left gaps will be filled using the auxiliary passive microwave product. Here the step two is crucial, and will be underlined in the paper.*

For more modifications, please see the new manuscript. In a word, we have adjusted and revised the manuscript following each your comments. Obviously, after these modifications the manuscript is more readable and easier to understand. Thank you again!

With respect to suggestion 3, please see the following detailed responses.

Minor comments and suggestions:

1. Line 48-55: People who are unfamiliar with MODIS snow products may be difficult to understand your introduction on these products. I would suggest you give a clearer description.

Response: Thanks for your suggestion. We have revised this description. Please see the sentences "*The National Snow and Ice Data Center (NSIDC) routinely produces and continually updates the standard MODIS snow products. Before the C6 version, there were only two sets of standard snow products — MOD10A1 and MYD10A1, which provide conventional SCE information just under clear skies. Since there are abundant cloud-induced gaps in the products, they in fact can not give the complete SCE knowledge. This is an obvious flaw for which reason the previous standard products were criticized most (Liang et al., 2008). As such, among the latest C6.1*

*version that is released very recently another two sets of new cloud-gap-filled (CGF) products, MOD10A1F and MYD10A1F, are introduced and generated (Hall et al., 2019). But as of now, this update has not been completed, and these products are only available in some years.*" in the new manuscript.

2. Section 2.1 line 88-93: You should introduce the MOD09GA, MYD09GA and MCD12Q1 concisely.

Response: Done. Thanks. Please see the new paragraph in section 2.1.

*MODIS products we use as the input data to generate new SCE data include: MOD09GA, MYD09GA, and MCD12Q1. MOD09GA and MYD09GA are the standard land surface reflectance products that are derived from Terra MODIS and Aqua MODIS, respectively, after the so-called atmospheric correction. They provide us the 500-m land surface reflectance from MODIS band 1 to band 7, as well as the mask information (e.g., cloud and water masks), and are our main inputting data. MCD12Q1 is the Terra\Aqua composite land-cover-type product, providing us the annual land-cover information that is generated according to the International Geosphere Biosphere Program (IGBP) land cover classification system. In the study, it is another important input which is used to indicate the detailed land cover types. For all of the three products, the newest C6.1version is adopted.*

3. Section 2.2: See the first suggestion. A simple introduction on the OLI snow maps is definitely needed.

Response: Done. Thanks.

4. Section 3. 3.1.1 line 130-138: Preliminarily screening, it is repeat for the last two

graphs.

Response: They appear similar but are different. We have reorganized them. Thanks!

*As mentioned just, the purpose of the preliminarily screening is to preclude the pixels that are impossibly covered by snow completely. Snow has the distinct spectral characteristic relative to other common land cover types. Generally, its reflectance is high in the visible spectrum, but rapidly drops in the infrared spectrum. As done by the standard MODIS snow products (Riggs et al., 2006), we can use the combination of MODIS band 2 and 4 within the visible spectrum, and band 6 within the infrared spectrum to preliminarily screen out the pixels that must be snow-free, but keep all possible snow pixels (even with a very low possibility) for a further discrimination.*

*For that purpose, we investigate all available snow samples, and find for Terra-MODIS more than 99% of the snow samples are constrained in the condition of band 2 ≥ 0.15, band 4 ≥ 0.05, and band 6 ≤ 0.45 . Therefore, the preliminarily screening rule of the Terra-MODIS is adjusted into: all possible snow pixels must meet the condition of band 2 ≥ 0.15, band 4 ≥ 0.05, and band 6 ≤ 0.45, and pixels that do not meet will be identified as snow-free immediately. Similarly, for Aqua-MODIS 99% of the snow samples are constrained in the condition of band 2 ≥ 0.12, band 4 ≥ 0.07, and band 6 ≤ 0.40 . The preliminarily screening rule of the Aqua-MODIS is set into: all possible snow pixels should meet this condition, and pixels that do not meet will be deemed as snow-free immediately.*

5. Section 3.1.2 line 154: Optimized NDSI thresholds, "However, as expected, only using the NDIS criterion seems not accurate enough to discriminate snows over

those forest land-cover types, except the "Evergreen Needleleaf Forest" (due to its sparse distributions in China)." Change "NDIS" to "NDSI"!

Response: Sorry, there is an obvious clerical error. We have corrected it!

6. Section 3.1.4: Postprocessing based on surface temperature and DEM, how to determine the threshold of surface temperature screen?

Response: The threshold of surface temperature is determined according to our previous investigation (Hao et al., 2021). For lowlands of DEM < 1300 m snow at the surface basically impossibly exists when their temperature is ≥ 275 K (2 degrees Celsius); but for highlands of DEM ≥ 1300 m, a higher temperature threshold, 281 K (8 degrees Celsius), seems more appropriate due to possible existences of warm snow on highlands.

Hao, X. H., Huang, G. H., Che, T., Ji, W. Z., Sun, X. L., Zhao, Q., Zhao, H. Y., Wang, J., Li, H. Y., and Yang, Q.: The NIEER AVHRR snow cover extent product over China – a long-term daily snow record for regional climate research, Earth Syst. Sci. Data, 13, 4711–4726, doi:10.5194/essd-13-4711-2021, 2021.

7. On Figure 2, please provide the full name of Figure 2.

Response: Done. Thanks.

8. Section 3.3.2, line 230: "For the aggregated SCE data"-> "For the aggregated SCE".

Response: Done. Thanks.

9. Section 4.1: Confuse matrix is a commonly-used tool to evaluate the products relevant to classes. It seems this section is unnecessary.

Response: Thanks for your advice. But if the confuse matrix is deleted here, it is really difficult to understand table 4, 5 and 6. More importantly, here a metric termed as "bias" is introduced, which is not a commonly-used index in the literature. Therefore, we keep the confuse matrix, but delete the headline of "4.1 Accuracy metrics".

10. Section 4.3, line 268-270: This may be attributed to different snow/non-snow number distributions in nature among different years, and varying sample numbers caused by different ground measurements available in different years. I cannot understand "different snow/non-snow number distributions in nature among different years".

Response: Thanks. We have deleted these words.

11. Section 5.2, line 325-335: for the two examples, are they all covered by forest?

Response: Yes, they are all covered by forest. The first one is about 20km*20km, and the second one is about 50km*20km.

12. Line 355: During our validations or comparisons, we found this phenomenon is somewhat common in the edges of snow-cover areas and the forest areas of Northeast China. Very awkward sentence. Please consider revising it.

Response: This sentence really seems abrupt here because there is not a background introduction before. Therefore, we have deleted it. Thanks for your advice.

13. Section 6, line 345: finally, a totally cloud-free SCE is mapped through replacing the residual gaps with auxiliary passive microwave snow-depth data. Is "finally the residual gaps are all filled according to the implication given by a auxiliary

passive microwave snow-depth dataset" better?

Response: Done. Thanks.

14. Section 6, line 350: "by a series of processes filling cloud-induced gaps". It seems

wordy here because you just mention them in the above paragraph!

Response: We have deleted these words. Thanks.

---

## Author Comment (AC3)

This study proposes a new MODIS snow-cover-extent product over China. The optimal NDSI thresholds varying with land cover types are extracted, the NDVI-NDFSI decision rules specific for snow discrimination in forest areas are optimized, and an HMRF gap-filling technique, which can simultaneously assimilate temporally and spatially neighbouring information, is imported. The need for such an approach is well justified and the authors cite ample relevant literature. The study provides examples demonstrating the successful performance of the method. The paper is basically well-written and presented.

There are a few important and minor comments/mistakes that are listed below and should be taken into account.

Response:Thanks for your positive comments. We have revised the manuscript following your suggestions or comments listed below. Please see the detailed responses.

Line 38, "on the other hand", where is "on the one hand".

Response: We have changed "on the other hand" into "in addition". Thanks.

Line 53, "the MODIS band 4 (0.55 μm) and band 6 (1.6 μm) reflectance" should be "the reflectance of MODIS band 4 (0.55 μm) and band 6 (1.6 μm)".

Response: Very good suggestion. We have revised this sentence. Thank you.

Line 54, "distinguish snow cover or not" should delete "or not".

Response: Done. Thanks.

Line 58, "research" should be "studies".

Response:According to suggestion 7 of referee # 1, we have revised these paragraphs. In the new manuscript, this sentence has been deleted. Thanks.

Line 78, the use of GEE is not motivated from the above text.

Response:All of our work is conducted on the GEE platform. Although this is not one of the key points of our paper, it is necessary to give this information here as a background introduction. Of course, our work can be done without GEE, but GEE indeed provides us much convenience. Therefore, we do not explain why we use GEE and only mention GEE as one piece of background information.

Line 80, "surface cover" should be "land cover".

Response:Done. Thanks.

Line 88, "chiefly" should be deleted.

Response:Done. Thanks. Besides, we have revised section 2.1 following the suggestion of referee # 1.

Line 88-91, all of the definite articles before the acronyms should be deleted. All of the "us" also need to be deleted.

Response:Done. Thanks. Please see the modified section 2.1.

*MODIS products we use as the input data to generate new SCE data include: MOD09GA, MYD09GA, and MCD12Q1. MOD09GA and MYD09GA are the standard land surface reflectance products that are derived from Terra MODIS and Aqua MODIS, respectively, after the so-called atmospheric correction. They provide the 500-m land surface reflectance from MODIS band 1 to band 7, as well as the mask information (e.g., cloud and water masks), and are our main inputting data. MCD12Q1 is the Terra\Aqua composite land-cover-type product, providing the annual land-cover information that is generated according to the International Geosphere Biosphere Program (IGBP) land cover classification system. In the study, it is another important input which is used to indicate the detailed land cover types. For all of the three products, the newest C6.1version is adopted.*

Line 102, only the snow-depth product has linkage?

Response: As mentioned in the manuscript, the other products are accessible directly at GEE. Therefore, we do not give their linkages.

Line 106, "too" should be deleted.

Response: Done. Thanks.

Line 108-115, why was snow-depth data used to validate SCE product? It is not a snow depth product. This need explanation in the text.

Response:Done. Please see the last sentence of section 2.4. At each station, snow-cover condition is determined by the criterion proposed by Klein and Barnett (2003). That is, if measured snow-depth is ≥ 1-cm, it is covered by snow; else it will be snow-free.

Line 118 and 234, "paper" should be changed to "study".

Response:Done. Thanks.

Line 119, "Sect." should be "Sec." or "Section".

Response:Done. Thanks.

Line 154, "NDIS" should be "NDSI".

Response:Done. Thanks.

Line 167, "≥" should be "greater than".

Response:There are many signs of "≥", "=" , "≤" in the manuscript. We use them because they can convey the same meaning but with less words. This also increases the readability, we think.

Line 187-191, I suggest to move Subsection 3.2 to the front of Subsection 3.1, and to combine Subsection 3.2 and 3.1. The authors should give a general introduction of the method first, and then present the details. There is the same problem in Subsection 3.4 and 3.3.

Response:Very good suggestion. Here we have reorganized these contents following the second general comment from referee # 2. In the revised manuscript, section 3 focuses on the algorithm, and the introduction to the new product is given in section 4. In the section 3, we first present the flowchart of the algorithm. Please see the new manuscript.

Line 281, "this section" should be changed to "we".

Response:Done. Thanks.

Table 3, 4, 5, and 6, OE is just one minus PA and CE is just one minus UA, thus OE and CE are not need to show if you use PA and UA, and PA and UA are not need to show if you use OE and CE.

Response:Yes, OE = 1- PA and CE = 1-UA. They are easy to understand, but we often mention only OE and CE in the text. We are afraid some readers may confuse if we do not present them in the tables straightforward.

Line 338 and 340, I suggest to delete "Under the support of several national programs of China" and "Toward this".

Response:Done. Thanks.

Line 345, "and finally" should be change to "finally".

Response:Done. Thanks.

Figure 1, 2, 3, 4, and 6 use "non-snow", but "snow-free" is used in the text and Figure 7.

Response:We think the word of "snow-free" may be better. But in the figure 1, 2, 3, 4, and 6, the spaces are very narrow. If the word of "snow-free" is used, it will overlay the important contents in the figures. Therefore, we used the word of "non-snow" to save space.

Figure 3, "Number density colorbar" should be "Number density".

Response:Done. Thanks. Please see the new figure below.

[Figure]

The parentheses in the caption of the figures should be deleted.

Response:We have deleted the parenthesis in the caption of figure 2 in the new manuscript, namely previous figure 1. As for figure 4 and 6, we think the contents in the parentheses are very important. If deleting them, it may decrease the readability of the figures.

---

## Author Comment (AC4)

This manuscript develops a novel method of creating seamless snow cover extent (SCE) data for China. The novelty of this study is significant in that there is currently no other dataset of SCE with such a long time series and quality (e.g., a full spatiotemporal continuity at 500m and daily resolutions, sufficient evaluations, and accuracy) that can be comparable with here developed. The improvement of performance in forested areas is also impressive. I have some minor concerns as follows.

Response:Many thanks for your positive comments. We have revised the manuscript following your suggestions or comments listed below.

- Landsat-8 OLI data are used as training data in this study, however, how were they processed as reference SCE data? By manually vectorizing or also using an NDSI-like method? This process of course has uncertainties that should be mentioned.

Response:Thanks for your constructive suggestion. In our study, Landsat-8 OLI snow products are derived using the improved "SNOMAP" algorithm developed by Chen et al. (2020). We have added this information in the new manuscript in section 2.2. It is an NDSI-like method. To ensure these products' accuracy, we also check them again by our visual interpretation. On average, their over accuracies are larger than 99% comparing to the results of our visual interpretation. On the spatial scale of 30-m, most pixels are pure snow or snow-free, and mixed pixels are infrequent. Therefore, a simple NDSI-like method can give a very high accuracy. Following your suggestions and those of referee # 1, we rewritten this section. Please see the newest section 2.2.

*To refine the decision rules (will be elaborated in section 3), we must obtain the quality MOD09GA and MYD09GA training samples in advance. Toward this, two groups of Landsat-8 Operational Land Imager (OLI) snow maps across China during the 2013 – 2018 snow seasons (beginning on Nov. 1st through Mar. 31st of the next year) are produced here. The first group is derived from 1509 scenes of OLI images, which will be regarded as "true" values to acquire the Terra-MODIS training samples; and the second group comes from 1648 scenes of OLI images, which will be used to acquire the Aqua-MODIS training samples. These snow maps are generated by the improved "SNOMAP" algorithm developed by Chen et al. (2020), and have a spatial resolution of 30-m. In every OLI snow map, there are only three classes — snow, snow-free, and cloud.*

*Then 30-m snow maps will be aggregated within the 500-m spatial window to indicate whether the corresponding MOD09GA and MYD09GA pixel is covered by snow or not. Within a spatially and temporally (in the same day) collocated MOD09GA/MYD09GA pixel, if no less than 50% of OLI pixels are snow-covered, then it is a snow sample. If over 50% of OLI pixels are snow-free, it is a snow-free sample. If the   most of the OLI pixels are the class of cloud, it will be deemed as an invalid sample and is subsequently discarded. Of course, all of these must be done under the condition that the collocated the MOD09GA/MYD09GA pixel is a clear pixel.*

*Finally, for the MOD09GA, totally 21.20 million snow samples and 17.66 million snow-free samples are obtained; for the MYD09GA, 12.05 million snow samples and 12.65 million snow-free samples are obtained in total. Note that it is necessary to obtain*

*the Terra-MODIS and Aqua-MODIS training samples separately, as MYD09GA band 6 is not the directly observing data (many sensor's detectors of this band have broken since the Aqua launch), but the restored data using the algorithm of Wang et al. (2006).*

Chen, S., Wang, X., Guo, H., Xie, P., Wang, J., and Hao, X.: A Conditional Probability Interpolation Method Based on a Space-Time Cube for MODIS Snow Cover Products Gap Filling, Remote Sens., 12, 3577, doi: 10.3390/rs12213577, 2020.

- Why were two groups of Landsat-8 snow maps used for training Terra and Aqua separately? I do not think the overpass time is a good reason for explaining this because most Landsat-8 data are in the morning which is closer to Terra. This could also be an uncertainty for training Aqua data.

Response:In fact, many Landsat-8 snow maps are same in the two groups. Of course, some are inconsistent because there may be a very different weather condition for Terra and Aqua overpasses. As you have noticed, Landsat-8 overpass time is closer to that of Terra. Therefore, they more likely have the consistent weather condition. If we use the same one group of Landsat-8 snow maps, the sample number of Aqua will much less than that of Terra. To ensure the sample number even, we have to import more Landsat-8 snow maps for Aqua. This will result in two groups of Landsat-8 snow maps are needed. Because Landsat-8 overpass time is closer to Terra's, Terra's training samples should be more accurate. Thanks.

- I am a bit confused by Table 2 because there are several duplicate land cover types in the first column and some types with multiple rows. It can be improved to be clearer.

Response:Thanks for your comment. We have split this table into two in the new manuscript. One is for Terra-MODIS, the other is for Aqua-MODIS. Indeed, the previous presentation is misleading. Please see the new manuscript.

- In section 2.4, I recommend adding more explanations for the use of station observations. I can understand that the station data were used as they were totally free of the effects of cloud blockage. However, for some readers, this could be confusing when Landsat8 snow maps were already used as reference.

Response:Done. Thanks for your suggestion. Please see the below description in the new section 2.4.

*Daily ground snow-depth observations up to 362 stations from the China Meteorological Administration (CMA) since 2000 will be used to validate or assess all associated MODIS SCE products* in *the study. To ensure the qualities of the measurements, most CMA stations obey the following observing specifications: 1) snow-depth is measured manually in an open spot near the station using a professional ruler; 2) the measurements were conducted only when the fractional snow cover in the field of view is larger than 50%; 3) all observations were done at 8:00 Beijing time every morning.*

*At each station, the surface true condition, snow-cover or snow-free, is determined by the criterion proposed by Klein and Barnett (2003). That is, if measured snow-depth is ≥ 1-cm, it is covered by snow; else it will be snow-free. Because ground measurements are not*

*limited by weather conditions, they are a better choice to independently validate all satellite-based SCE products.*

- For section 3.1.4, the use of ERA5 LST could be a (possibly not big) problem or not optimal. The MODIS standard product uses MODIS LST with a high resolution of 1km in the aid for temperature screening, however, ERA5 only has a coarse resolution of 0.25° which is much larger than the pixel size of MODIS SCE data.

Response:Yes, maybe MODIS 31 brightness temperature is a better choice, as done by MODIS standard snow products. But unfortunately neither MOD09GA/MYD09GA nor GEE provide this data. As you may notice, it is a very huge workload to download and process the MODIS 1B products for that data. Here the quality of LST may be not that important because the temperature screening is only for deleting a few obviously false snow pixels. Thanks.

- In Figure 7, the microwave-based snow depth data are suggested to be additionally plotted against other data. It can be seen in Figure 7 that there are large data gaps in both Terra and Aqua snow map on that day and the microwave-based snow depth data could provide valuable reference data of snow in those areas.

Response:Thanks for your suggestion. The purpose of Figure 7 is to present an example of our product, and demonstrate what new SCE datasets look like. The microwave-based snow depth data is not our product. It is really weird to put them together because it is irrelevant to our product. Besides, the spatial resolution of microwave-based snow depth

data is so coarse that the figure may look very ugly due to many rectangular spots. From the figure, one can intuitively see after our gap-filling there is no gap existing in the new CGF SCE dataset, but clear-sky SCE datasets are affected by clouds severely. This is the reason why we give Figure 7. Here we do not want to demonstrate the microwave-based snow depth data is promising because it is not affected by clouds.

- The English writing has a large potential to be improved.

Response:Thanks for your comment. We know as a non-native speaker there must be language problems throughout the manuscript, more or less. Therefore, before the submission to HESS it in fact had been edited by a professional company, EDITSPRINGS. Please see the below certificate (at that time, our manuscript title is "The NIEER MODIS snow extent product over China"). After this revision, we asked them to polish the English again, and then check our English carefully again. I hope the new manuscript is much better than the previos.

[Figure]

[Figure]

**EDITORIAL CERTIFICATE**

This document certifies that the manuscript listed below was edited for proper English language, grammar, punctuation, spelling, and overall style by one or more of the highly qualified native English speaking editors at EditSprings.

Manuscript title:
The NIEER MODIS snow cover extent product over China

Authors:
Hao Xiaohua, Huang Guanghui, Zheng Zhaojun, Sun Xingliang, Ji Wenzheng, Zhao Hongyu, Wang Jian, Li Hongyi, Wang Xiaoyan

Date Issued:
Aug 27 2021

Certificate Number:
ES-202106241645578090

This certificate can be verified on www.editsprings.com/query.asp. This document certifies that the manuscript listed above was edited for proper English language, grammar, punctuation, spelling, and overall style by one or more of the highly qualified native English speaking editors at EditSprings. Neither the research content nor the authors' intentions were altered in any way during the editing process. Documents receiving this certification should be English-ready for publication; however, the author has the ability to accept or reject our suggestions and changes.

Typo errors:

Lines 65-66: please add a reference

Response:Done. Thanks.

Zhang, H. B., Zhang, F., Zhang, G. Q., Che, T., Yan, W., Ye, M., and Ma, N.: Ground-based evaluation of MODIS snow cover product V6 across China: Implications for the selection of NDSI threshold, Sci. Total Environ., 651, 2712-2726, doi:10.1016/j.scitotenv.2018.10.128, 2019.

Lines 133-138: the two paragraphs could be combined to reduce duplications

Response:Done. Thanks. Please the new manuscript.

*For that purpose, we investigate all available snow samples, and find for Terra-MODIS more than 99% of the snow samples are constrained in the condition of band 2 $\geq$ 0.15, band 4 $\geq$ 0.05, and band 6 $\leq$ 0.45 . Therefore, the preliminarily screening rule of the Terra-MODIS is adjusted into: all possible snow pixels must meet the condition of band 2 $\geq$ 0.15, band 4 $\geq$ 0.05, and band 6 $\leq$ 0.45, and pixels that do not meet will be identified as snow-free immediately. Similarly, for Aqua-MODIS 99% of the snow samples are constrained in the condition of band 2 $\geq$ 0.12, band 4 $\geq$ 0.07, and band 6 $\leq$ 0.40 . The preliminarily screening rule of the Aqua-MODIS is set into: all possible snow pixels should meet this condition, and pixels that do not meet will be deemed as snow-free immediately.*

Line 154: NDIS should be NDSI

Response:Very good suggestion! Thanks.

Line 213: gas should be gap?

Response:Very good suggestion! Thanks.

Line 300. Terra? I guess you'd like to say Aqua?

Response:Very good suggestion! Thanks.

---

## Author Response (AR2)

**Editor comments to the author:**

The authors need to do some technical corrections, before final accepted.

**Response:** Thanks for your decision. We have done these technical corrections following the comments of the three referees.

**Report #1**

The paper was considerably improved from its earlier version. The authors complied with all recommendations from four reviewers. The paper is recommended for publication.

**Response:** Thank you very much.

**Report #2**

Thank the authors for improving the manuscript according to the reviewers' comments. It can be accepted subject to some technical corrections, e.g., in Line 267, "table 4" should be "Table 4"; in the equations of Table 4, the use of "T" is not needed. You can directly use "SS + SN + NS + NN" as the denominator.

**Response:** Thanks for your careful review for our manuscript. We have gone through the manuscript again and corrected the problems you had pointed out.

**Report #3**

I am satisfied with the revisions. The revised manuscript is suggested to be accepted. I only have a minor suggestion. At line 69, I do not think Zhang et al. 2019 is a good citation to be used here as their work is not related with the contents.

**Response:** Thanks for your constructive suggestion. Indeed, this citation seems inappropriate. We have changed it into that of Hall et al. (1995). In their paper, they wrote "Furthermore, if there are any inaccuracies in the input products, that is, inaccuracy in the calculation of the atmosphere correction, this may cause errors in the amount of snow " in page 133, which indicate the importance of the atmosphere correction.

Hall, D. K., Riggs, G. A., and Salomonson, V. V.: Development of Methods for Mapping Global Snow Cover Using Moderate Resolution Imaging Spectroradiometer Data, Remote Sens Environ, 54, 127-140, Doi 10.1016/0034-4257(95)00137-P, 1995.

Thanks again!